# Expectation violations enhance neuronal encoding of sensory information in mouse primary visual cortex

Matthew F. Tang [1,2,3,8] ✉, Ehsan Kheradpezhouh[1,2,8], Conrad C. Y. Lee[1,2,4], J. Edwin Dickinson [5], Jason B. Mattingley [2,3,6,7] & Ehsan Arabzadeh [1,2]

The response of cortical neurons to sensory stimuli is shaped both by past events (adaptation) and the expectation of future events (prediction). Here we employed a visual stimulus paradigm with different levels of predictability to characterise how expectation influences orientation selectivity in the primary visual cortex (V1) of male mice. We recorded neuronal activity using two-photon calcium imaging (GCaMP6f) while animals viewed sequences of grating stimuli which either varied randomly in their orientations or rotated predictably with occasional transitions to an unexpected orientation. For single neurons and the population, there was significant enhancement in the gain of orientation-selective responses to unexpected gratings. This gain-enhancement for unexpected stimuli was prominent in both awake and anaesthetised mice. We implemented a computational model to demonstrate how trial-to-trial variability in neuronal responses were best characterised when adaptation and expectation effects were combined.

There is often more information in the sensory environment than the brain has the capacity to fully process. To cope with this information overload, activity within neuronal circuits is modulated by processes such as adaptation[1,2] attention[3,4], and prediction[5,6]. Neural adaptation is known to improve the transmission of sensory information in circuits by accounting for the statistics of past sensory inputs[1,7,8]. Likewise, selective attention can enhance neural responses to task-relevant features and suppress irrelevant information[3,9]. An influential theory of neural function argues that predictions about specific future stimuli, based upon Bayesian inference, might similarly improve the fidelity of stimulus representations[5,6]. Based on this predictive coding view, the mammalian cortex is conceptualised as a predictive machine that uses the statistical regularities of incoming sensory inputs to iteratively generate an internal model of its external environment. Predictive coding provides a simple theoretical view of perception which is supported by a substantial body of work in human neuroimaging and

behavioural studies[10,11]. The classic mismatch negativity effect has become a hallmark of this literature[12,13]. When encountering an unexpected stimulus, the brain generates a significantly larger evoked response compared with the response following an expected stimulus[11]. Decoding of activity from electroencephalography (EEG) recordings in humans has revealed that expectation affects the representation of visual information in the brain[14–18].

Recent work supports the idea that prediction influences single neuron responses across a number of sensory modalities[19–24]. Theoretical models propose that higher level processing regions generate inhibitory copies of the expected stimulus which are passed down the cortical hierarchy to the earlier processing regions[6], where they are integrated with incoming sensory inputs. If a stimulus is expected, the inhibitory copy should minimise the neuronal response. By contrast, any mismatch between the expected and presented stimulus should result in a prominent response.

[1]Eccles Institute of Neuroscience, John Curtin School of Medical Research, The Australian National University, Canberra, ACT, Australia. [2]Australian Research Council Centre of Excellence for Integrative Brain Function, Victoria, Australia. [3]Queensland Brain Institute, The University of Queensland, Brisbane, QLD, Australia. [4]School of Biomedical Sciences, The University of Melbourne, Melbourne, VIC, Australia. [5]School of Psychological Sciences, The University of Western Australia, Perth, WA, Australia. [6]School of Psychology, The University of Queensland, Brisbane, QLD, Australia. [7]Canadian Institute for Advanced Research (CIFAR), Toronto, Canada. [8]These authors contributed equally: Matthew F. Tang, Ehsan Kheradpezhouh. ✉e-mail: matthew.tang@anu.edu.au

Here, we tested key elements of predictive coding theory at the neuronal level in mouse primary visual cortex (V1). We used two-photon calcium imaging in awake mice that were exposed to sequences of oriented gratings at different levels of predictability. We characterised how prediction affects orientation selectivity in V1 neurons, and how changes in orientation tuning modulate the amount of information about the sensory input carried by individual neurons and neuronal populations. We demonstrate that unexpected stimuli significantly increase the gain of orientation selectivity without any corresponding changes to the width of the tuning function. Such increased gain to expectation violations yields increased information about stimulus features within single-cells and at the level of neuronal populations. This enhanced representation of unexpected stimuli is present in both awake and anaesthetised mice. Finally, we use a computational model to quantify the contribution of adaptation and expectation to neuronal responses at the single trial level.

## Results

We combined experimental and modelling approaches to determine how prediction affects neuronal responses in mouse (C57BL) V1 cortical neurons to sequences of oriented grating stimuli. We asked whether the selectivity of individual neurons changes with expectations about the orientation of future stimuli by presenting sequences of gratings with different levels of predictability to awake mice ($N = 3$ across 23 sessions in total, 1693 neurons) while imaging Layer 2/3 activity in V1 using two-photon excitation microscopy (Fig. 1a–c, Supplementary Movie 1). The stimulus sequence was adapted from the Allen Brain Institute's Brain Observatory paradigm[25] used to quantify orientation selectivity. Each sequence consisted of a series of full-screen gratings (0.034 c/°, 50% contrast) oriented between 0° and 150° in 30° steps, presented at 4 Hz with no inter-stimulus interval. In the Random condition (Fig. 1b, c), the orientations of successive gratings were uncorrelated.

To establish predictions about stimulus orientation, in the Rotating condition the grating rotated either clockwise or anti-clockwise

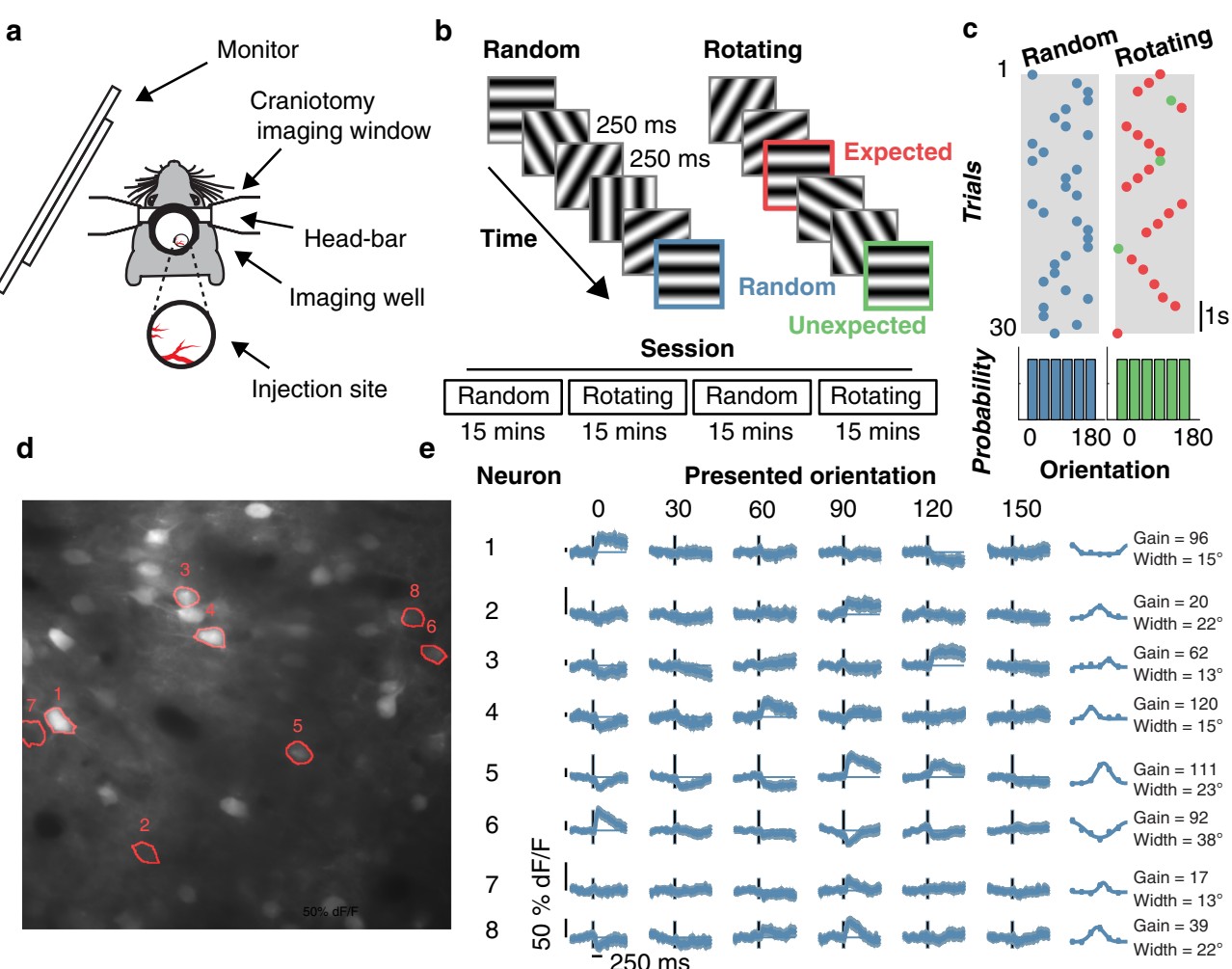

**Fig. 1 | Experimental procedure for testing the effects of prediction on orientation selectivity in mouse V1 neurons. a** Apparatus for using two-photon calcium imaging in combination with visual stimulation. **b** Schematic of the Random and Rotating sequences of oriented gratings. **c** In the Random condition, the orientation of each stimulus was drawn from a pseudo-randomised distribution (uniform probability from 0 to 150° in 30° steps). In the Rotating condition, the gratings rotated clockwise (e.g., 0° -> 30° -> 60°) or anti-clockwise (e.g., 0° -> 150° -> 120°) for 5–9 presentations (red dots) before jumping to a random unexpected orientation (indicated by the green dots). **d** Mean motion-corrected two-photon image from a single session, with individual neurons highlighted in red. **e** Time course of activity in the corresponding neurons highlighted in **d** in response to different grating orientations from the Random condition. The tuning functions in the right panels show the average response from 0 to 1000 ms after stimulus presentation. Points are fitted with a circular Gaussian with a baseline offset. The key parameters of the fits are given as the gain (amplitude) and width (standard deviation) of the Gaussian for each neuron. Shading and error bars show ±1 standard error over trials.

for 5–9 presentations (in 30° steps), before jumping to an unexpected random orientation. In this condition, Expected events were those which constituted the rotating sequence, whereas Unexpected events were those in which the stimulus jumped randomly to an unpredicted orientation. Critically, for unexpected events the jump from the predicted orientation was to a random orientation matched to the correlation statistics for the stimulus sequence embedded in the Random condition. Figure 1b, c identify the three types of transitions within the visual stimulation protocol: Random transitions (in blue), Expected transitions (in red) and Unexpected transitions (in green). Figure 1d, e shows eight example neurons imaged within a field of view, each of which exhibited a varying degree of orientation selectivity under the Random condition. In line with previous work[25], many imaged neurons showed orientation selectivity for the spatial frequency employed (462/1693; one-way ANOVA $p < 0.05$ for orientation selectivity).

## Prediction affects single neuron activity

We next examined how orientation selectivity of individual neurons was affected by stimulus predictability (Fig. 2). The three example neurons shown in Fig. 2a all exhibit orientation selectivity from ~85 to 100 ms after stimulus onset. The first neuron (top row of Fig. 2a) responded maximally to gratings at 0°, with slight suppression for the more distant orientations (60°, 90°, 120°). During presentation of the Expected stimulus (red trace), modulation of neuronal activity began before the onset of the stimulus (0 ms). This pre-stimulus modulation is due to the rotating nature of the sequence: the stimulus presented at −500 ms was orthogonal to that presented at 0 ms. This means that in the 0° condition, the anti-preferred stimulus (90°) was presented at −500 ms, whereas in the 90° condition, the preferred stimulus (0°) was presented at −500 ms. The rotating nature of the stimuli during the Expected sequence thus produced an idiosyncratic temporal profile in neuronal response. For this reason, here we focus on the Random and Unexpected transitions where the stimuli presented immediately before 0 ms were uncorrelated with the current stimulus.

The main effect of predictability is evident from the three example neurons illustrated in Fig. 2a. There was a systematic increase in neuronal responses to the preferred orientation, and a decrease to the anti-preferred orientation, in the Unexpected (green trace) compared with the Random condition (blue trace). This response profile is consistent with a positive gain modulation for unexpected gratings. The overall population response (aligned to the preferred orientation) showed the same pattern of results (Fig. 2b), with an increased response to the preferred stimulus in the Unexpected versus Random condition. The responses of 133/462 orientation-selective neurons (28.8%) were significantly modulated in the Unexpected condition relative to the Random condition (t-test, $p < 0.05$). Of these, all but two (98.5%) showed a larger response in the Unexpected condition (Fig. 2d), and this increase in selectivity emerged shortly after stimulus presentation (Fig. 2e).

We next quantified how orientation selectivity was affected by predictability. To do this, we fitted circular Gaussian tuning functions to separately determine the gain (amplitude) and width (standard deviation) parameters of orientation selectivity for each neuron (Fig. 2f, Supplementary Fig. 1, see Eq. 1). The gain of the tuning curve was significantly greater in the Unexpected condition than in the Random condition (t(461) = 15.67, $p < 0.001$). By contrast, there was no difference in the width between these two conditions, (t(461) = 1.58, $p = 0.12$, Supplementary Fig. 1). These results are consistent with our recent work examining how prediction affects orientation selectivity measured non-invasively in humans[14,15]. A control condition showed these effects were not due to the systematic rotations that followed Unexpected gratings (Supplementary Fig. 2).

## Prediction affects population coding of orientation

In our initial set of analyses, we found that expectation affected orientation selectivity in individual V1 neurons. We next examined how enhanced orientation selectivity for unexpected stimuli at the single-neuron level in turn shaped the information contained within the population response. Previous human neuroimaging studies using multivariate pattern analysis have shown that expectation affects classification accuracy of the stimulus features[14–17,26]. To determine how these findings generalise across species, we applied a similar multivariate pattern analysis to the neuronal population data. We used all imaged neurons ($N = 1693$; 23 imaging sessions), including both orientation-selective and non-orientation selective neurons to decode the presented orientation using inverted/forward encoding modelling (see multivariate analysis section in Methods for details). Figure 3a, b illustrates the key steps in a forward (or inverted) encoding approach and how this method can be used to determine the amount of orientation-selective information contained in the population activity on a trial-to-trial basis. In line with the human work[14–17,26], in a first step the method applies an encoding model using a subset of trials (training trials) to estimate neuronal selectivity to each orientation (Fig. 3a). Then, in a second step, it inverts these weights to reconstruct the stimulus representation from the population response on a new set of test trials (Fig. 3b).

We first applied this decoding procedure in a time-resolved manner to determine the temporal dynamics of population-level prediction effects (Fig. 3d). This showed the decoding performance started to rise for the Random and Unexpected conditions shortly after stimulus presentation. More importantly, greater decoding accuracy emerges for Unexpected relative to Random stimuli from shortly after stimulus onset (~100 ms). The early divergence suggests that the increase in selectivity for unexpected stimuli results from expectations developed before the stimulus appears rather than from a subsequent top-down influence which would appear later. Unsurprisingly, in the Expected condition orientation information could be decoded above chance before the stimulus appeared. This is because orientations occurring before stimulus presentation (0 ms) were correlated with the orientation of the decoded stimulus presented at time zero. The decoding profile for Expected stimuli also exhibits an oscillating profile, which likely reflects a combination of three factors: oscillations in neuronal activity due to the periodic onsets of stimuli in the presented sequences; the 30° changes in orientation from one stimulus to the next within the rotating sequences; and the dynamics of the calcium indicator.

We next examined the effect of different-sized neuronal populations on decoding accuracy (Fig. 3e). To do this, we selected groups of neurons and used a 10-fold cross-validation procedure to train and test the classifier, which was repeated 24 times with different subsets of neurons. The decoding procedure was performed on the average neuronal responses from 250 to 1000 ms after stimulus onset, and different-sized pools of neurons were selected (1 to 1600 neurons, in 100 logarithmically-spaced steps). This analysis again showed that the presented orientation was decoded significantly better in the Unexpected than the Random condition. Figure 3e illustrates that this effect emerged with population sizes of relatively few neurons (<10). The Expected condition also showed greater decoding accuracy relative to the Random condition, but this effect was smaller than in the Unexpected condition and did not emerge until a population of ~100 neurons was included in the analysis.

## Predictions repel perception away from the expected orientation

The analyses presented above reveal a higher gain in orientation selectivity among V1 neurons following Unexpected grating stimuli relative to otherwise identical gratings within Random sequences. According to formal models of predictive coding, the magnitude of a

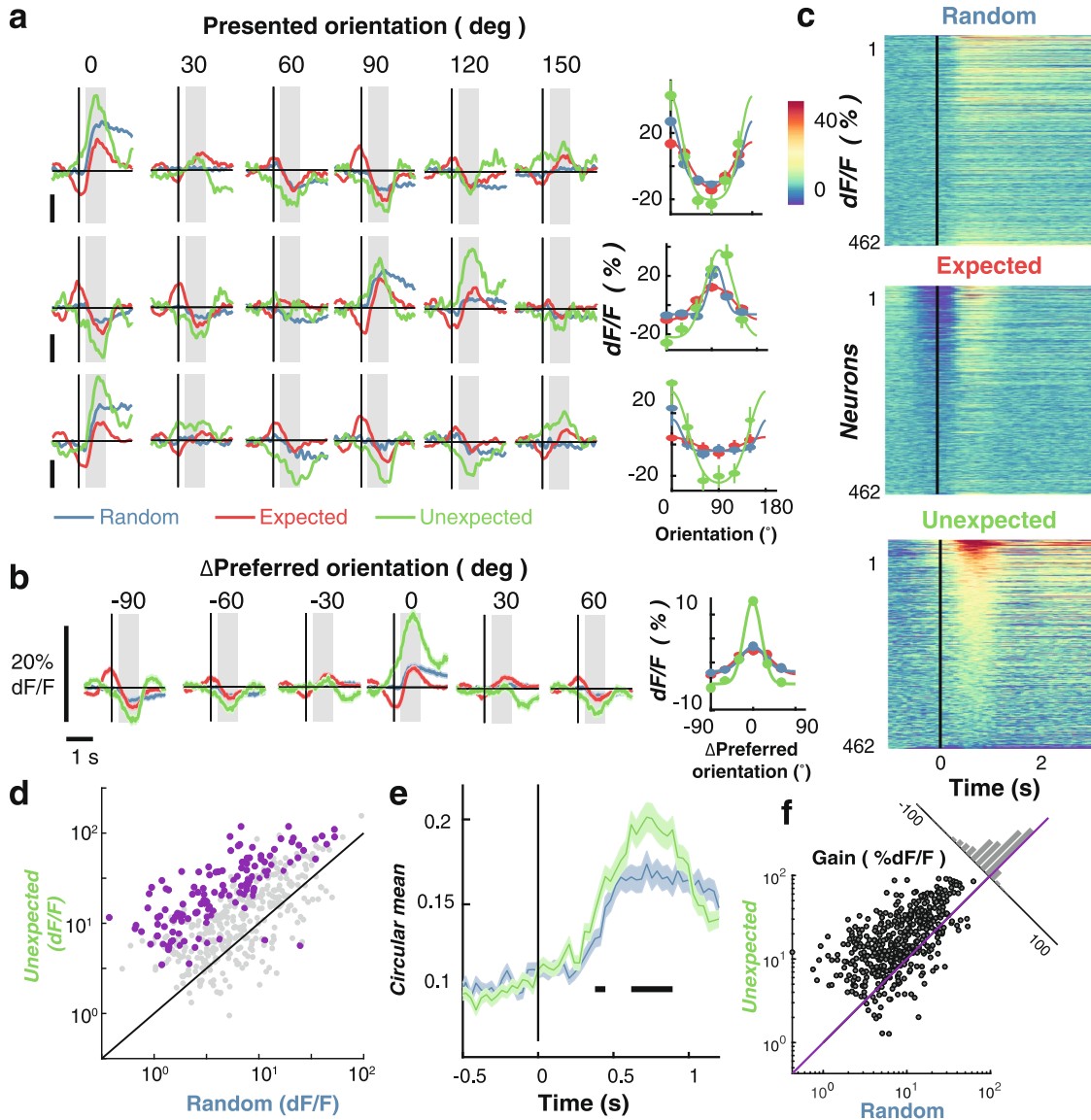

**Fig. 2 | Expectation affects orientation-selective responses of individual V1 neurons. a** Time-courses of three example neurons in response to oriented grating stimuli in the expected, Unexpected and Random conditions. Each neuron is illustrated in a separate row, with the rightmost panel showing orientation tuning curves for that neuron. The tuning is measured as the averaged response from 250 to 1000 ms after stimulus onset (grey shading). The solid curve is a fitted Gaussian function with a constant offset. **b** Same as in A, but shows activity for all orientation-selective neurons ($N = 462$ neurons) aligned to their preferred orientation (0°) to allow averaging. Right panel: Same as in A but showing the Gaussian tuning function for the population response. **c** Response to the preferred orientation across the three conditions for all orientation-selective neurons. For presentation the time-courses are smoothed with a Gaussian with a 33.3 ms kernel. Every row represents the response of one neuron. In each panel, neurons are sorted based on their evoked response in the Unexpected condition (most excited on the top).

**d** Comparison of the response in the Unexpected and Random conditions at the preferred orientation. Each dot represents one neuron. Purple dots show neurons significantly modulated by expectation ($N = 133$ neurons); grey dots are non-modulated neurons ($N = 329$ neurons). **e** Time-course of orientation-selectivity (circular mean) for the Random (blue) and Unexpected (green) conditions. Black horizontal lines indicate timepoints with statistically significant difference between conditions, determined using non-parametric cluster-corrected procedures (see Methods). **f** Summary statistics ($n = 462$) for fitted Gaussian parameters across the population for the different sequence types. All parameters are shown in Supplementary Fig. 1 for all three conditions. The Gain is the amplitude of the Gaussian. The insert shows the distribution of the difference between the two conditions (random minus unexpected). The purple line shows the zero point. Across all panels error bars and shading represent ± 1 standard error of mean. All statistical tests were two sided.

prediction error should be determined by the degree of surprise, with more surprising stimuli yielding larger responses[5,6]. Consistent with these models, we have shown in human observers that orientation-selective stimulus-evoked responses increase as the difference between expected and presented stimuli also increases[15].

In the current study, we were able to quantify the degree of prediction error in the Rotating condition and use this index to characterise any change in orientation-selective responses in individual V1 neurons. To do this, we grouped orientation-selective neurons ($N = 462$) based on their maximum orientation-selective

response in the Random condition (Fig. 4a and Supplementary Fig. 3). We found that orientation selectivity was influenced by the expected orientation, such that responses were smallest when the expected orientation was closest to the preferred orientation. For example, as shown in Fig. 4a, neurons tuned to 90° had the lowest orientation tuning when a 90° grating was expected (darkest green line). Orientation selectivity was reduced to a lesser degree when the surrounding orientations (60° and 120°) were expected, suggesting that the magnitude of the prediction error affected neuronal responses in an orientation-selective manner.

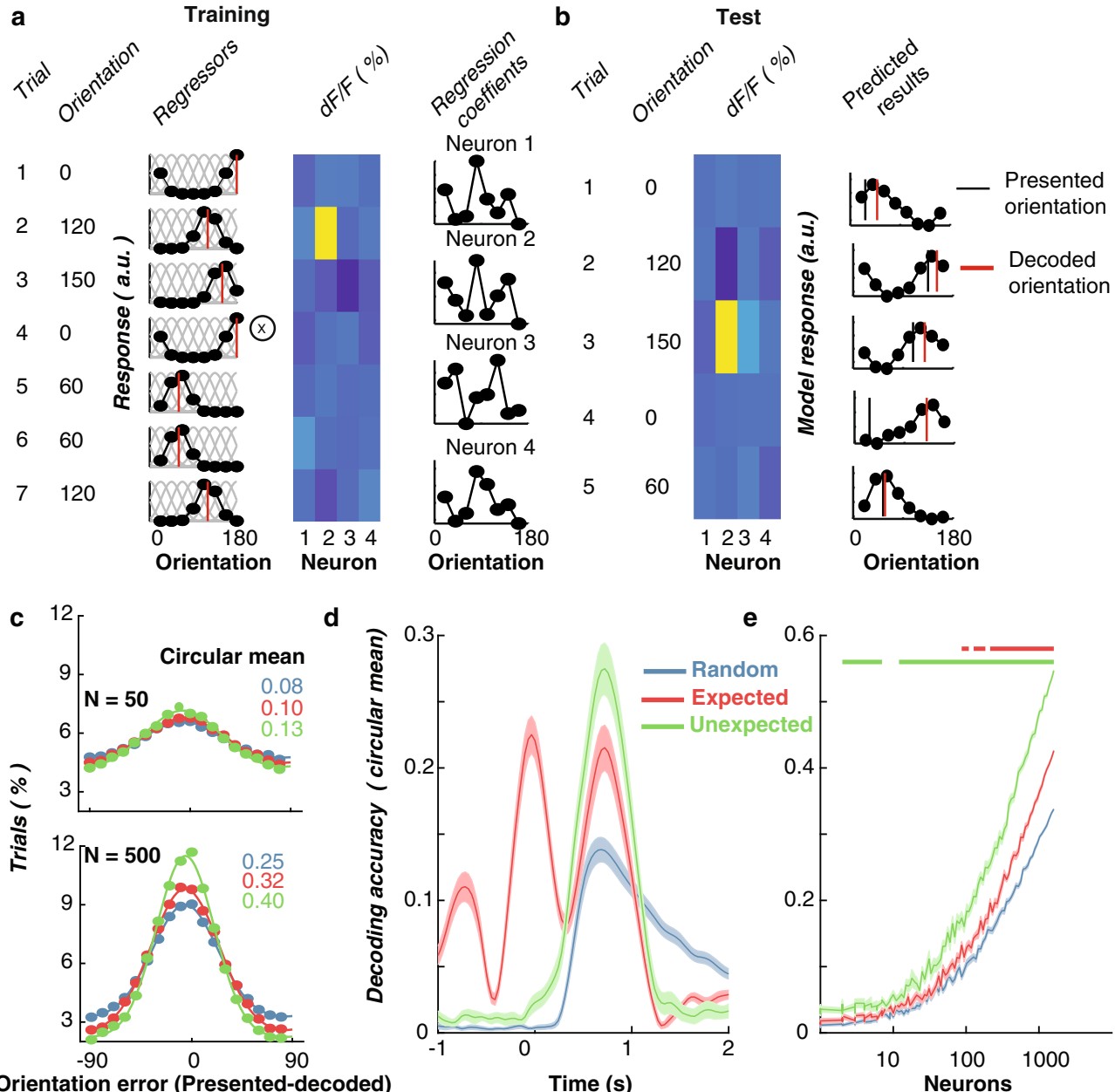

**Fig. 3 | Expectation affects stimulus-specific information carried by neuronal population activity. a** Schematic of training the multivariate forward orientation encoding. Example regressors for 7 training trials with different orientations. The basis functions (grey lines) in response to different orientations which produce the regressor weights. Neuronal response for four example neurons for the example trials. Least squared regression is applied between the regressors and response to determine selectivity. Regression coefficients (beta weights) for four example neurons for each of the regressors found from a training set of data. **b** Testing the encoding model. Activity for the four neurons in test trials. Inverting the regressor weights and multiplying them by the population responses from the four neurons produces the predicted orientation response from this pattern of activity. The difference between the predicted and presented orientation for a given stimulus is the orientation error. **c** Distribution of orientation error when encoding was performed separately on groups of 50 neurons and 500 neurons at a time (with 24 permutations of different neuronal combinations). The vector average of these histograms was taken as the decoding accuracy for each condition. The coloured numbers show the vector sum for the corresponding curves. **d** Time-resolved classification from forward encoding modelling ($N = 500$ neurons) with 24 permutations of different groups of neurons. **e** Decoding accuracy scales with the number of neurons. The classifier was trained and tested on the average response from 250 to 1000 ms following stimulus onset, with different numbers of neurons included ($N = 24$ permutations of different neurons for each population size). The coloured horizontal lines indicate statistical significance using sign-flipped cluster permutation tests comparing Random vs. Unexpected (green line) and Random vs. Expected (blue line). In panels **d** and **e**, shading/error bars indicate ±1 standard error of the mean across permutations.

To better visualise these effects, we aligned all neurons to their preferred orientation and replotted the data as a function of the difference between the expected orientation and the preferred orientation (Fig. 4b). To quantify these effects, we fit Gaussian curves to each neuron's orientation selectivity for all expected orientations (Fig. 4c, d). Both the gain (Fig. 4c, one-way ANOVA,

$F_{(5,1835)} = 3.31$, $p = 0.006$, $\eta^2 = 0.006$) and the baseline response to all orientations (Fig. 4d, $F_{(5,1835)} = 8.38$, $p < 0.001$, $\eta^2 = 0.022$) were systematically affected by the magnitude of the violated expectation.

We followed up these results by examining how population-level encoding of the presented orientation was affected by the magnitude

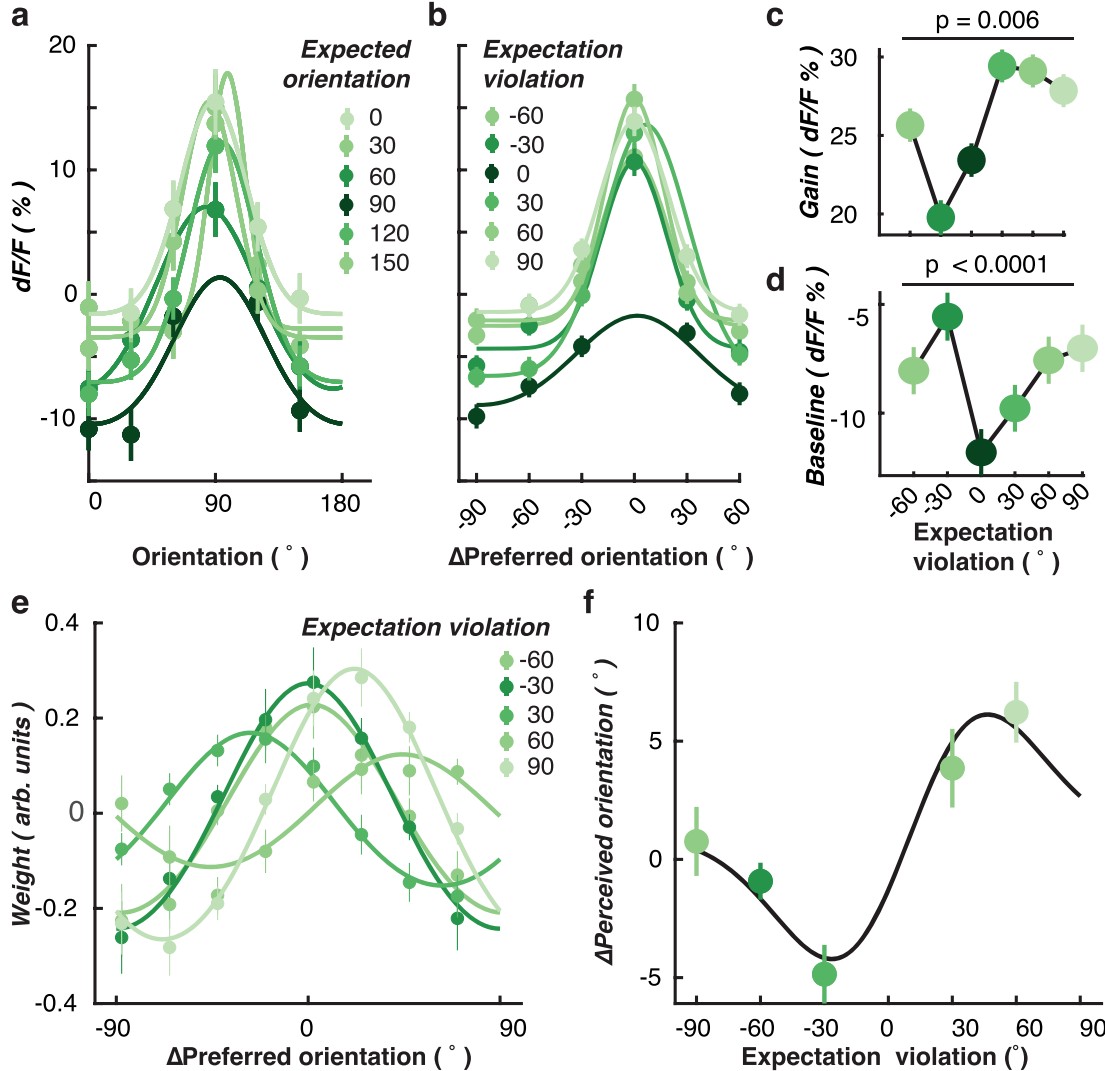

**Fig. 4 | Increase in neuronal responses to unexpected stimuli is determined by the magnitude of the prediction error. a** Neurons tuned to each displayed orientation are affected differently when different orientations are expected. Panel **a** shows an example for each expected orientation using neurons selective for 90° orientations (*n* = 92 neurons), as defined based on their responses in the Random condition (from 250 to 1000 ms). Responses of remaining neurons selective for the other presented orientations are shown in Supplementary Fig. 3. For each unexpected stimulus in the rotating condition, we identified the difference between the orientation of the expected stimulus and the orientation of the presented unexpected stimulus. For instance, if 60° was expected but 0° was unexpectedly presented, the expectation violation would be 60°. **b** All orientation-selective neurons (*n* = 462) aligned with their preferred orientation, plotted as separate Gaussians for each difference between the expected orientation and the presented orientation (expectation violation). **c** Gain and **d** baseline of Gaussians fitted to each neuron's response (*n* = 462), plotted as a function of expectation violation for all orientation-selective neurons. **e** Forward encoding modelling reveals how population representations of orientation are affected by the degree of expectation violation. The encoding weights are shown separately here for different values of expectation violation. Encoding was performed on population response recorded in each session (*n* = 23 sessions). **f** The *y* axis shows the difference between the presented and decoded orientation (ΔPerceived orientation). The population response (filled symbols) is biased away from the expected orientation with the largest bias at ±30° (*n* = 23 sessions). In all panels, error bars indicate ±1 standard error of the mean. All statistical tests were two sided.

of the prediction error (or expectation violation). To do this, we divided the forward encoding results (Fig. 3) into separate bins based on the difference between the expected and presented orientation (Fig. 4e). We found that the decoded orientation of the presented stimulus was biased away from the expected orientation, with the largest effect observed for a difference of 30°. This "repulsion" effect is reminiscent of the well-known adaptation aftereffect for oriented stimuli[27,28], in which the largest effect typically arises when the adapting and test stimuli are separated by around 30°. In the present experiment, however, the observed repulsion effect was driven by the expected orientation rather than the orientation of the preceding stimulus.

## Computational modelling of the relative contributions of adaptation and prediction on visual coding efficiency

Formal models of predictive coding assume that high-level cortical areas pass predictions, which are inverse copies of the expected stimulus, to lower-level areas[5,6]. According to this framework, only a small neuronal response is required for representation if a stimulus matches the expectation[29]. Such an account is reminiscent of the effect of adaptation on neuronal representation, whereby an immediately preceding stimulus reduces the neuronal response to a current stimulus without decreasing the overall amount of stimulus information[1,8]. Indeed, a number of studies have investigated whether adaptation might be due to prediction errors[14,30,31]. Both adaptation and prediction

rely on the statistics of sensory inputs. Adaptation exploits the recent history of stimulus presentations to alter current sensory representations, whereas prediction is thought to use statistical regularities to extract future patterns.

We created a simple computational model of orientation processing to better understand how expectation interacts with adaptation to influence the neural coding of orientation. The model is based on several tuned orientation-selective neurons (or information channels) maximally sensitive to different orientations. The neurons respond proportionally based on their sensitivity to the presented orientation (Fig. 5). We incorporated two sources of inhibition: adaptation (in response to a previously presented stimulus) and expectation (in response to a predicted future stimulus). Similar to previous work[27,28,32,33], adaptation causes gain modulation in neuronal orientation selectivity based on the response to the preceding stimulus (Fig. 5a, b). Prediction, on the other hand, affects neuronal responses by producing an inverse copy of the expected orientation. To account for commonly observed long-lasting effects of gain modulation on orientation sensitivity[34,35], the model allows sensitivity to recover gradually over a number of trials. The amount of gain modulation can be varied to increase or decrease the influence of either adaptation or prediction.

We presented sequences of orientations to the model from both the Random and Rotating conditions to determine whether it can explain the observed changes in orientation selectivity at the single-trial level. Because there are two sources of gain (adaptation and expectation), we assume an equilibrium of gain modulation is available to the system to allow it to maintain population homeostasis[36]. To this end, in the initial model we implemented 0.5 arbitrary units (arb. units). of gain available, which was varied in the two stimulus conditions. In the Random condition, the adaptation gain was set to 0.5 (arb. units) and the expectation gain was set to 0 arb. units because the stimulus sequence was completely unpredictable. In the Rotating condition, by contrast, the gain for both expectation and adaptation were set to 0.25 arb. units. We re-aligned neurons (Fig. 5d) to their preferred orientation and determined their response to stimuli under different conditions by fitting the same Gaussian to the results (Fig. 5e, f). Consistent with the neuronal data (Fig. 2), in the model the gain of orientation selectivity increased in the Unexpected condition ($M = 0.64$, SD = 0.05) relative to the Expected ($M = 0.59$, SD = 0.03) and Random ($M = 0.55$, SD = 0.02) conditions. The Unexpected trials resulted in greater orientation selectivity than the Expected trials because sensitivity to the stimulus was reduced for a different orientation (the predicted grating orientation) than the one that was presented (Fig. 5e). As with the experimental data, the width of tuning was similar for the Unexpected ($M = 29.8$, SD = 0.62) and Random ($M = 30.06$, SD = 0.59) conditions, whereas the Expected condition was slightly wider ($M = 32.18$, SD = 0.79, Fig. 5f). The model produced a qualitative fit consistent with the effects of expectation on V1 orientation selectivity. The modulation of stimulus selectivity is consistent with previous work which found that uncommon stimuli result in increased stimulus-specific adaptation in auditory cortex[37], and that the V1 population response adapts to high-level stimulus statistics in a homeostatic manner[36].

We next determined whether the model provided a quantitative fit to the recorded neuronal activity. To do this, we used the model to generate predictions about neuronal responses, which we regressed against the actual data for each neuron. Specifically, for each experimental session for the awake mice, we presented the model with the same orientation sequence viewed by the mouse, which in turn generated a predicted response for each simulated neuron on every trial. We used ridge regression to determine beta weights for each of the six regressors (orientations) for the three different gain settings for each neuron.

We found that a greater proportion of the variance in the trial-to-trial activity of neurons could be explained when the model

incorporated inhibition from expectation (Fig. 5i). We presented the orientation sequences from the Rotating condition to the model with three different gain responses for expectation. With no gain, only the presented stimulus determined the response of the model. As gain was increased from 0.25 and 0.75 arb. units, greater inhibition from expectation increased the model's fit with the data (Fig. 5i).

For the adaptation model, there was no significant increase in its ability to explain neuronal activity with increasing gain (Greenhouse-Geisser corrected; one-way ANOVA, $F(1.87, 420.23) = 0.66, p = 0.62$). By contrast, the explanatory power of the expectation-only model greatly increased with increasing levels of gain (one-way ANOVA, $F(2.04, 458.98) = 21.87, p < 0.001$). Furthermore, the model that incorporated a moderate amount of adaptation (0.25) with varying degrees of expectation gain best predicted the neuronal response (one-way ANOVA, $F(1.79, 403.06) = 30.55, p < 0.001$). A 3 (Model type; Adaptation, Expectation, Combined model) × 5 (Gain level; 0,0.2,0.4,0.6,0.8,1.0) repeated-measures ANOVA confirmed this observation, revealing that both the type of model ($F(1.78,404.61) = 23.71, p < 0.001$), and the gain level ($F(2.03,456.78) = 35.00, p < 0.001$) significantly affected the proportion of variance explained. These factors significantly interacted ($F(1.99) = 17.87, p < 0.001$), showing that the difference in explanatory power between the models increased with increasing gain. Follow-up tests showed that the expectation model and combined model explained significantly more variance than the adaptation model across all gain levels (Bonferroni correct $p < 0.001$) but neither explained more than one another (Bonferroni correct $p = 1.00$).

### Predictive coding under anaesthesia

Finally, we asked whether global anaesthesia altered the influence of prediction on orientation selectivity observed in awake animals. Previous work in humans on expectation violations has reported larger neural responses to unexpected than to expected stimuli during sleep[38,39], in different attention states[15,40], and when individuals were in a coma[41], vegetative state[42–44] or under anaesthesia[45]. These findings suggest that the influence of prediction errors on patterns of brain activity varies across different global brain states and levels of consciousness. To address this issue at the level of individual V1 neurons, we conducted a further experiment in which the stimulus sequences (Random versus Rotating) were displayed to mice under urethane anaesthesia ($n = 3$ animals). Before each recording session, the mouse was anesthetized by intraperitoneal administration of urethane/chlorprothixene (0.8 g/kg and 5 mg/kg body weight, respectively). All other methodological details were identical to those described for the awake recordings. For each mouse, we ran the full stimulus protocol with 2–4 different areas in V1 (11 in total, 576 neurons). We found 96/576 (16.6%) neurons were orientation selective. As shown in Fig. 6a, b, the gain of orientation selectivity was again significantly enhanced in the Unexpected relative to the Random condition (Fig. 6c, $t(95) = 5.64$, $p < 0.0001$). As in awake animals, there was a small but non-significant decrease in the width of the tuning curve in the Unexpected condition relative to the Random condition Supplementary Fig. 5, $t(95) = 0.39, p = 0.70$).

Finally, for each neuron we calculated the "surprise" effect by subtracting the gain of the Gaussian tuning curve for the Unexpected condition from that of the Random condition (Fig. 6d). A value larger than 0 indicates that the neuron's orientation selectivity was enhanced in the Unexpected condition. There was no significant difference in the magnitude of the surprise effect in awake animals compared with those that had been anaesthetised ($t(556) = 1.38, p = 0.17$), suggesting that the influence of prediction errors on orientation-selective responses in V1 neurons is equivalent for awake and anaesthetised animals.

### Discussion

Here we provided an experimental test of how neuronal representations of visual information are affected by prediction in the primary

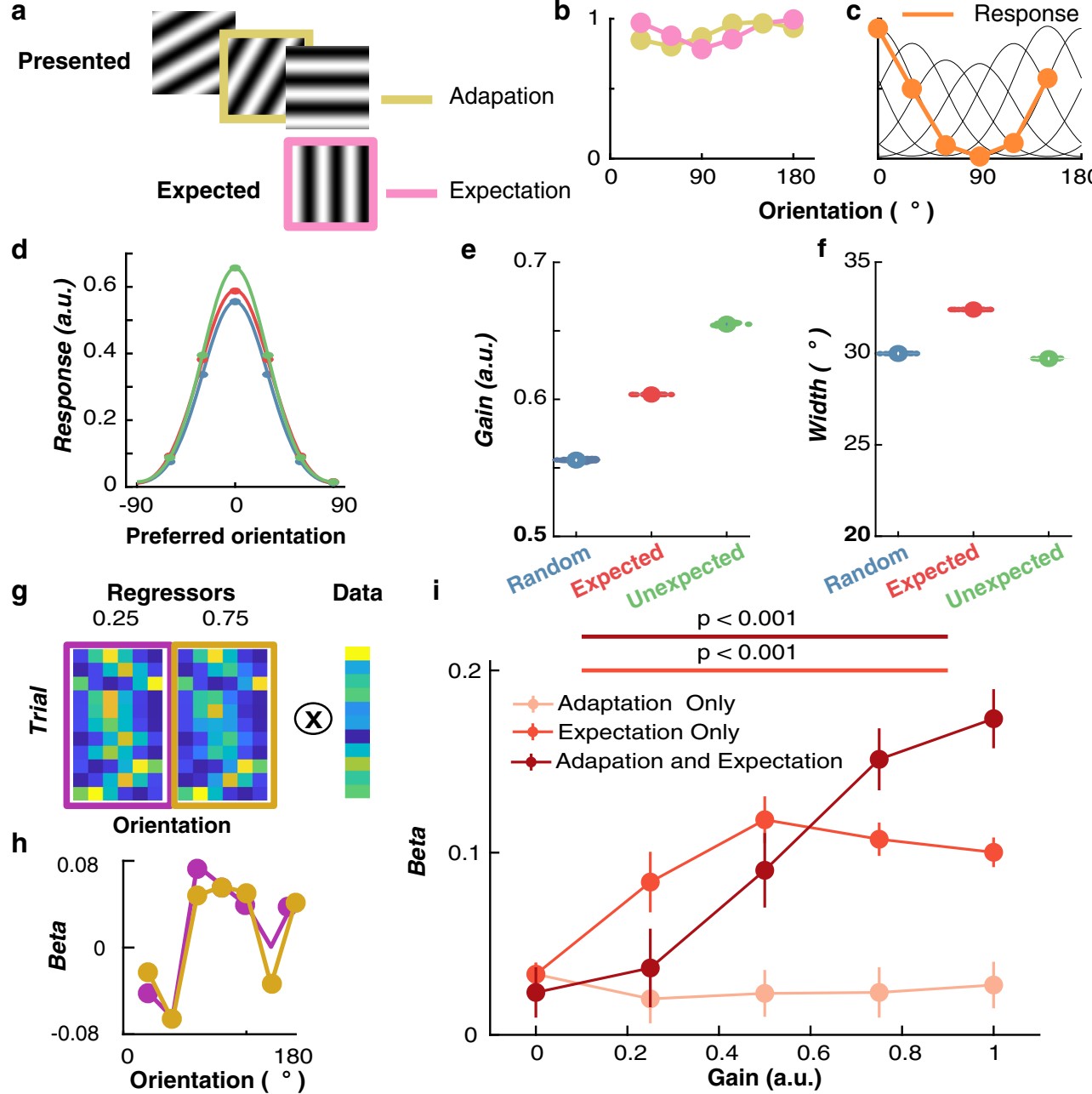

**Fig. 5 | Computational model for explaining variance in neuronal response by incorporating gain modulation from prediction and adaptation effects.** The model consists of a bank of six neurons maximally selective for different orientations. The model's sensitivity is affected by previous orientations in the sequence (Adaptation) as well as future predicted orientations (Expectation). These factors determine the response to the presented orientation on each trial. **a** An example sequence of trials in the rotating condition. The orientations of the preceding (mustard) and expected (pink) trials determine the adaptation and the expectation gains. **b** The adaptation gain (mustard line) is determined by the orientation of the previous stimuli. The expectation gain (pink line) is determined by the inverse copy of the response to the expected orientation. **c** Collectively, the two gains modulate the sensitivity of the channels on the next trial. These weights for the different orientations are applied to the model's sensitivity channels (black lines), which give the response (orange line) to the presented orientation (vertical dashed line; in this case 0°). **d** Dots indicate the responses of the channels, and the curves are fitted

Gaussian functions. Fitted parameter values to the model's responses for the different stimulus conditions showing gain (**e**) and width (**f**) of the response to each session ($n = 23$) data. The large dots show the median and the smaller dots show the session results. The error bars indicate the upper and lower quartile range. **g** An example testing which model parameters best match the neuronal response in mouse V1 neurons. Regressors for two different expectation gains (0.25 and 0.75) lead to slightly different weights for 10 example trials. Warmer colours indicate higher values. These are regressed against the response (dF/F%) of each neuron. **h** This yields beta values for each orientation channel (regressors) for the two different expectation gains. **i** Ridge regression results when the model was used to predict response to the stimulus in the Expected sequence, with different levels of modulation from adaptation and prediction. The regressor (orientation) with the highest beta weight was chosen for each neuron ($n = 226$; modulated by prediction regardless of whether they were orientation selective). Error bars indicate ±1 standard error of the mean. All statistical tests were two sided.

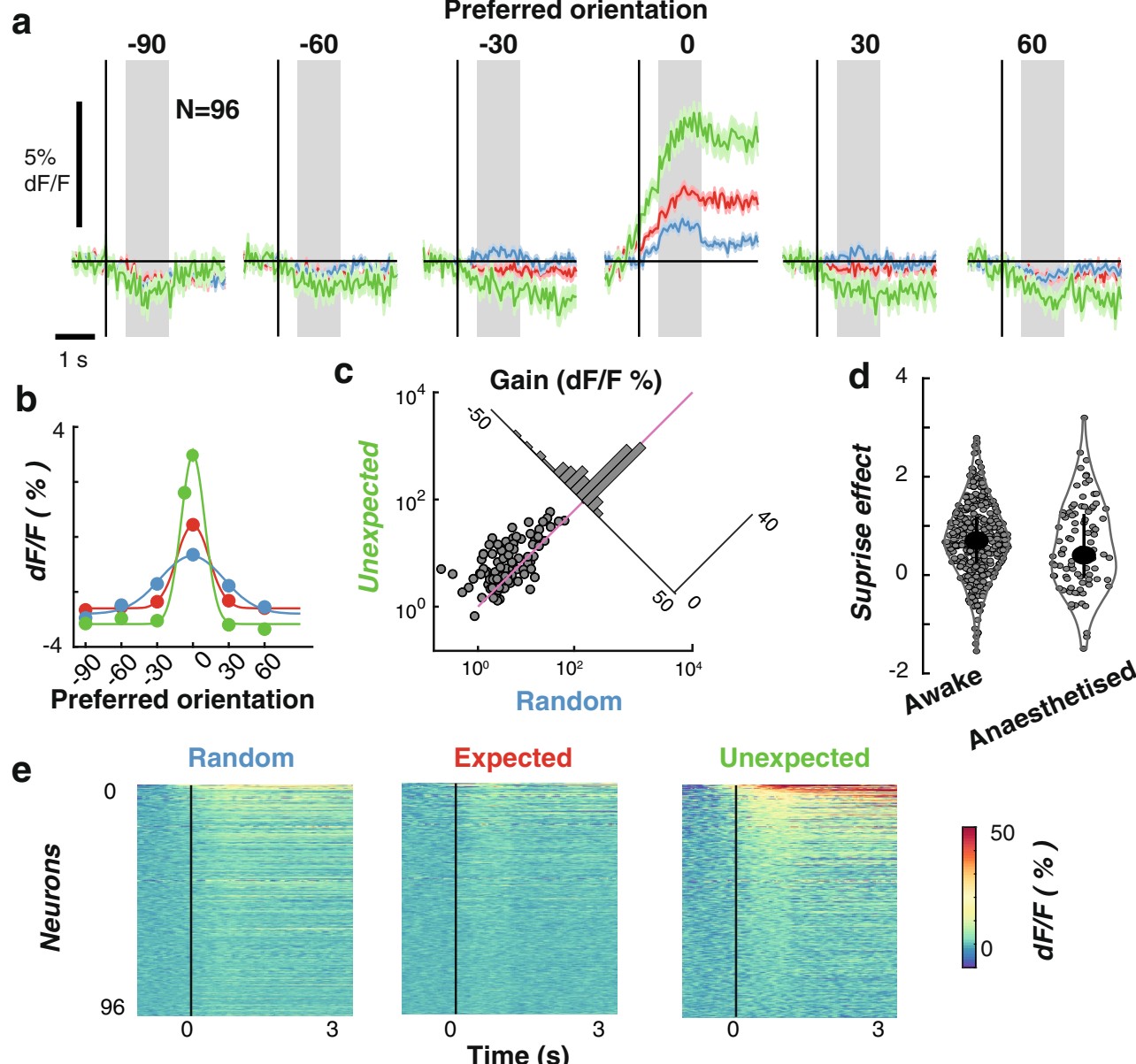

**Fig. 6 | Expectations affect the gain of orientation-selective V1 neurons under anaesthesia (N = 96). a** Time courses for all orientation-selective neurons aligned to their preferred orientation to allow averaging. Shading indicates ±1 standard error of the mean across neurons. **b** Population orientation tuning for the three expectation conditions, averaged across an epoch from 250 to 1000 ms after stimulus presentation. Solid lines are fitted Gaussian functions with a constant offset. **c** Summary statistics for the gain of the fitted Gaussians in **b**. The insert shows the distribution of the difference in response (Random minus Unexpected) for each neuron. The purple line shows the 0 point. **d** Comparison of the "surprise" effect (random events minus unexpected) between awake (N = 462) and anaesthetised (N = 96) experiments. The small dots are individual neurons and the larger points show the median. The error bars indicate the upper and lower quartile range. **e** Time course of responses to the preferred orientation of each neuron, shown separately for the three conditions. Neurons have been sorted by their responses in the Unexpected condition. Across panels **a–c** shading and error bars indicate ±1 standard error of the mean across neurons.

visual cortex (V1). Awake mice viewed streams of oriented grating stimuli in either a Random condition, in which there was no correlation between the orientations of successive stimuli (Random stimuli), or in a Rotating condition, in which grating orientations were mostly predictable from previous events within the sequence (Expected stimuli), but in which an occasional random orientation appeared unexpectedly (Unexpected stimuli). Expectations reliably modulated the gain of orientation selectivity in V1 activity, both at the level of single neurons and across the population of recorded neurons. We found that neurons tuned to an expected orientation showed a large decrease in their response compared with those not tuned to the expected orientation. The expectation violation response was also reliably present under

anaesthesia, suggesting that the relevant visual circuits utilise predictive patterns in the sensory input even when the animal is unconscious. Finally, we provided a computational implementation of a predictive coding model in V1 to better understand the interaction between adaptation and prediction. By varying the parameters of the model, we found that the best explanation for the observed neuronal activity relied on both inhibition from adaptation in response to immediately preceding stimulus events, and expectations about future stimulus features.

Our model of expectation-violation responses is phenomenological, in the sense that it describes our results in a way that is grounded in neuronal and synaptic mechanisms. This model contrasts

with more formal accounts based upon hierarchal predictive coding. Generally, neuronal responses to violations of expectations are formulated as precision-weighted prediction errors[45–53]. In other words, neuronal responses reflect the difference between sensory afferents and top-down predictions that are modulated or weighted by precision. Precision, in this context, is a prediction of predictability, as opposed to prediction of the sensory input. In the context of precision-weighted prediction errors, we can associate adaptation gain with the effects of predictability (i.e., precision weighting) and excitation gain with the prediction error per se; namely, the disinhibition of stimulus-bound responses by absent top-down predictions. This fits with predictive coding accounts of the mismatch negativity, where the equivalent effects are sometimes discussed in terms of stimulus-specific adaptation and a sensory memory component—which in turn correspond to predictions of precision and stimuli, respectively[14,15].

While the notion that predictions about the future affect perception was first proposed by Helmholtz[54], direct evidence in support of this idea at the level of individual sensory neurons has been lacking. A number of more recent theoretical models[5,6] have proposed a 'predictive coding' framework with the common idea that the brain inhibits sensory representations of expected stimuli to increase coding efficiency. Although there is good evidence that predictions affect the magnitude of neural responses measured with non-invasive, whole-brain neuroimaging methods[10,55], few studies have addressed whether individual neuronal responses are affected, even though this is a critical component of predictive coding models. The current results obtained from mouse V1 neurons fit well with our previous findings in humans, which suggest that orientation selectivity changes with expectation[14–16,26]. Specifically, and in line with the current work, forward encoding modelling of EEG activity revealed an increase in the gain, but not the width, of orientation tuning to unexpected stimuli in human observers[14].

Our results add to the understanding of how expectations affect the representation of sensory information. Previous work[19,22,23] has suggested that locomotion-induced increases in activity in primary visual cortex in mice relates to predictive coding[56,57] (but see ref. [57] for a different interpretation of these findings). Under the predictive coding framework, the increased activity caused by locomotion creates an expectation that the stimulus should move and change size. A prediction error is generated if the stimulus remains static, as is typical when measuring orientation selectivity, or moves in an inconsistent direction. There is significantly less locomotion-induced increase in response if the stimulus is made to move as the animal moves[23]. Our results are consistent with these findings, but also identify an enhanced gain mechanism reflected by a larger response to the neuron's preferred orientation.

In the human literature, expectation appears to affect sensory responses through different neural oscillatory frequency bands[58,59]. Recordings in macaques suggest visual information is fed forward through high-frequency gamma (60–80 Hz) oscillations, while feedback occurs through slow theta-band (14–18 Hz) activity[60,61]. As the present recordings were conducted using two-photon imaging with a relatively slow sampling rate, we are at present unable to determine the possible role of different oscillation frequencies in the observed expectation effects. Future work in which activity is recorded from multiple sites concurrently using electrophysiology could help characterise the distinct contributions of top-down and bottom-up neural signals to this expectation-induced gain modulation.

## Methods

### Mouse information

A total of five male wild-type mice (C57BL) were used for this study; two only awake, two only anaesthetized, one in both awake and anaesthetized. The mice were acquired from the Australian Phenomics Facility. All methods were performed in accordance with the protocol approved by the Animal Experimentation and Ethics Committee of the Australian National University (AEEC 2015/74; 2019/11). Mice were housed in a ventilated and air-filtered climate-controlled environment with a 12-h light–dark (8 am lights on, 8 pm lights off) cycle. The mice were kept in cages attached to Tecniplast Smart Flow. The system keeps the cage temperature at 22 degree C, but the humidity is not controlled but was typically around 40–50%. Mice had ad libitum access to food and water. Mice were transfected at 4–5 weeks with recordings starting 4–5 weeks later, and lasting between 2 and 3 weeks. Mice were culled at a maximum of 12 weeks of age. No statistical methods were used to calculate the sample size, but these were consistent or exceeded many other studies in the field.

### Expression of Ca$^{2+}$ indicator GCaMP6f

Mice were briefly anaesthetised with isoflurane (~2% by volume in O$_2$) in a chamber and moved to a thermal blanket (37 °C, Physitemp Instruments) before the head was secured in a stereotaxic frame (Stoelting, IL). Thereafter, the anaesthetic gas (isoflurane, ~2% by volume in O$_2$) was passively applied through the nose mask at a flow rate of 0.6–0.8 L/min. The level of anaesthesia was monitored by the respiratory rate, and hind paw and corneal reflexes. The eyes were covered with a thin layer of Viscotears liquid gel (Alcon, UK). The scalp was opened with ~5 mm rostrocaudal incision at the midline using scissors and the periosteum was gently removed. A circular craniotomy was made over the right visual cortex (3 mm diameter; centred 2 mm lateral and 4.5 mm posterior to Bregma) with the dura left intact. A glass pipette (15–25 μm diameter at tip) containing GCaMP6f (AAV1.Syn.GCaMP6f.WPRE.SV40, Penn Vector Core, The University of Pennsylvania, USA) was inserted into the cortex at a depth of 230–250 μm below the dura using a micromanipulator (MPC-200, Sutter Instruments, Novato, CA, USA). GCaMP6f was injected at 4–6 sites (with four 32-nL injections per site separated by 2–5 min; rate 92 nLs$^{-1}$). V1 was localised anatomically using coordinates established using functional methods[60]. Injections were centred 2 mm lateral and 4.5 mm posterior to Bregma. Injections were controlled using a Nanoject II injector (Drumont scientific, PA). After virus injection, the craniotomy was covered with a 3 mm diameter cover-glass (0.1 mm thickness, Warner Instruments, CT). This was glued to the bone surrounding the craniotomy. Custom-made head bars were fixed to the skull over Bregma using a thin layer of cyanoacrylate adhesive and dental acrylic. A small well was built surrounding the craniotomy window using dental acrylic to accommodate distilled water required for the immersion lens of the 2-photon microscope.

Ca$^{2+}$ imaging was performed using a two-photon microscope (Thorlabs Inc., Newton, NJ, USA) controlled by ThorImage OCT software (ThorImageLS, v3). The visual cortex was illuminated with a Ti:Sapphire fs-pulsed laser (Chameleon, Coherent Inc., Santa Clara, CA, USA) tuned at 920 nm. The laser was focused onto L2/3 cortex through a 16x water-immersion objective lens (0.8NA, Nikon), and Ca$^{2+}$ transients were obtained from neuronal populations at a resolution of 512 × 512 pixels (sampling rate, ~30 Hz). To abolish the effect of visual stimuli on the calcium signals, we filled the gap between the objective and the well with removable adhesive (Blu-Tack).

The obtained images were processed using the Suite2p toolbox (https://github.com/cortex-lab/Suite2P) for motion correction and segmentation. The surrounding neuropil signal was subtracted for each neuron's calcium traces. These corrected traces were high-pass filtered before the median response for each neuron was subtracted to determine dF/F.

### Visual stimulus

The stimuli were displayed on a 22-inch LED monitor (resolution 1920 × 1080 pixels, refresh rate 60 Hz) using the PsychToolbox presentation software for MATLAB[62,63]. The mouse was placed next to the monitor, which subtended 76.8° × 43.2° (one pixel = 2.4′ × 2.4′)

orientated 30° from their midline. The visual stimulus sequence was based on the Allen Brain Institute Brain Observatory paradigm used to measure orientation selectivity in mice. The stimuli were full-screen gratings (0.034 c/°, 50% contrast) displayed for 250 ms with no inter-stimulus blank interval giving a 4 Hz presentation rate. The spatial frequency was chosen to be close to optimal sensitivity of neurons in V1[25]. The orientations of the gratings were equally spaced between 0 and 150° in 30° steps so we could characterise each neuron's orientation selective profile.

The predictability of the orientations of the gratings was varied in the two stimulus conditions. In the random condition, the orientations of the gratings were drawn from a pseudo-random distribution with no relationship between the current orientation and the previous orientation. In the Rotating condition, the orientations of the gratings rotated (in 30° steps) either clockwise or anti-clockwise for 5–9 presentations, before jumping to an unexpected random orientation, where it began rotating in the opposite direction. The random and rotating conditions were presented in blocks of trials which were pseudorandomised in time within each imaging session.

In 3 mice, we ran a total of 23 imaging sessions and collected data from 1693 neurons. Neurons from all sessions and mice were pooled for analysis. One session (1.5–2 h) was recorded in a day from each mouse. These sessions occurred between 1 and 4 times per week. In each session, two runs of Rotating and Random sequences were presented, and each of these contained 1800 trials, alternating between Rotating and Random. The order of sequences was counter-balanced across mice. For some sessions for 2 of the mice, we also presented a rotating control condition to determine whether the systematic rotational movement after the unexpected jump affected orientation selectivity. In this condition, after the unexpected orientation the stimulus made another jump to a random orientation before starting to rotate in the opposite direction as the previous rotation. The number of events was increased from 7200 (3600 × 2) in each condition to 8400 to have the same number of unexpected trials as the original Rotating condition, while all other details remained identical with the Rotating condition. We ran 13 sessions in these two mice for all three conditions to compare the effect of the control. For all conditions, there was a balanced number of presentations of all the orientations.

## Data analysis

To determine the effect of predictability, we averaged the calcium response (dF/F%) from 250 to 1000 ms after stimulus presentation to derive tuning curves for each condition. To quantify how expectation affected the gain and selectivity of orientation-selective neurons we fitted circular Gaussian distributions with a constant offset (Eq. 1) using non-linear least square regression.

$$G(x) = A \exp -\frac{(x - \phi - j*180)^2}{2\sigma^2} + C \tag{1}$$

where $A$ is the gain (amplitude) of the Gaussian, $\phi$ is the preferred orientation of the neuron (in degrees), σ is the width (in degrees) and C a constant offset to allow for baseline shifts in the activity of the neuron. We searched for best fitting solutions with parameter j, with a search space from −4 to +4 in integer steps.

Neurons were selected for the primary analysis if they showed significant orientation selectivity (one-way ANOVA) in either the Random or Unexpected trials. To provide another test of how prediction affects orientation selectivity of individual neurons, we found the circular mean[64] of the averaged orientation tuning curve across all presentations within the condition (Fig. 2e). This was done for each time point (1/sample rate) between −500 and 2000 ms around stimulus presentation.

## Multivariate encoding analysis

We used a multivariate encoding approach (forward encoding modelling) to determine how the population activity carried information about the orientation of the presented grating on a trial-to-trial basis. This is adapted from human neuroimaging approaches, which examine orientation/feature selectivity from multivariate non-invasively recorded neural activity[14,15,65–68], but is similar to encoding approaches used to describe neuronal response to sensory stimuli[69,70]. Compared to the encoding-only, forward encoding takes the individual neuron activity to reconstruct the stimulus representation from the population activity. The technique goes beyond more commonly used multivariate pattern analysis procedures by producing tuning curves showing the full representation (in both gain, width, and bias) relative to the accuracy-only score.

The data were pooled across all experimental sessions with both orientation and non-orientation selective neurons used. In the first instance, we examined how the number of neurons affected decoding on a fixed time interval (250–1000 ms) and in the second instance, we found the time-resolved selectivity by applying the decoding procedure at each time point around the presentation of the stimulus (−500 to 2000 ms). A 20-fold cross-validation procedure was used in both instances for test and training data. The procedure evenly splits each test block to have the most even distribution of stimuli in each fold.

We used the presented orientations to construct a regression matrix with 8 regression coefficients. This regression matrix was convolved with a tuned set of nine basis functions (half cosine functions raised to the eighth power) centred from 0° to 160° in 20° steps. This helps pool similar orientations and reduces overfitting[70]. This tuned regression matrix was used to measure orientation information across trials. This was done by solving the linear Eq. 2:

$$B_1 = WC_1 \tag{2}$$

where $B_1$ (Neurons × N training trials) is the data for the training set, $C_1$ (8 channels × N training trials) is the tuned channel response across the training trials, and $W$ is the weight matrix for the sensors to be estimated (Neurons × 8 channels). We separately estimated the weights associated with each channel individually. $W$ was estimated using least square regression to solve Eq. 3:

$$W = (C_1 C_1^T)^{-1} C_1^T B_1 \tag{3}$$

We removed the correlations between neurons, as these add noise to the linear equation. To do this, we first estimated the noise correlation between neurons (which stops finding the true solution to the equation) and removed this component through regularisation by dividing the weights by the shrinkage matrix[68,71]. The channel response in the test set $C_2$ (8 channels × N test trials) was estimated using the weights in (4) and applied to activity in $B_2$ (Neurons × N test trials), as per Eq. 4:

$$C_2 = (W W^T) W^T B_2 \tag{4}$$

To avoid overfitting, we used 10-fold cross validation, where X-1 epochs were used to train the model, and this was then tested on the remaining (X) epoch. This process was repeated until all epochs had served as both test and training trials. We also repeated this procedure for each point in the epoch to determine time-resolved feature-selectivity. To re-align the trials with the exact presented orientation, we reconstructed the item representation by multiplying the channel weights (8 channels × time × trial) against the basis set (180 orientations × 8 channels). This resulted in an Orientation (−89° to 90°) × trial × time reconstruction.

To quantify the orientation selective response, we found the vector sum of the orientation for each trial (Fig. 3) to determine the decoded orientation. The difference between the decoded and

presented orientation was the orientation error. For each condition (and time point where applicable) we found the distribution of orientation errors and calculated the histogram of responses.

In the temporal classification analysis, groups of 500 neurons were used in each instance for both training and test data with the cross-validation procedure applied to each time point around stimulus presentation. We permuted new groups of 500 neurons 24 times. Next, we averaged the evoked activity from 250 to 1000 ms after stimulus presentation. To determine how decoding was affected by population size, the same classification was then used as in the previous analysis but with different numbers of neurons (2–1600 neurons in 100 logarithmically spaced steps). Again, we selected different groups of neurons 24 times so as not to skew the results by the neurons that were selected.

### Statistics
Non-parametric signed permutation tests[71,72] were used to determine time-resolved differences between conditions. The sign of the data was randomly flipped ($N = 5000$), with equal probability, to create a null distribution. Cluster-based permutation testing was used to correct for multiple comparisons over the timeseries, with a cluster-form threshold of $p < 0.05$ and significance threshold of $p < 0.05$. All statistical tests were two-sided and the alpha was set at 0.05.

### Computational model
The analytic model is based on previous work accounting for feature (i.e. orientation, spatial, colour) adaptation based on neuronal response and human psychophysical data[27,28,33,73,74]. The model consists of a bank of six orientation-selective information channels with preferred orientations evenly spaced between 0 and 150° (in 30° steps). Each channel's sensitivity profile is given by a Gaussian function (Eq. 5).

$$G(x) = A \exp - \frac{(x - \phi)^2}{2\sigma^2} \qquad (5)$$

where $A$ is the gain (amplitude) (set to 1 arb. units), $\phi$ is the channel's preferred orientation, $\sigma$ is the width of the channel (set to be 40° consistent with the neuronal data). The number of channels, along with the width means the model is equally sensitive to all orientations. The population response to any presented orientated stimulus is given by the sensitivity profiles of the channels (See Supplementary Fig. 4). In an unadapted state (Supplementary Fig. 4A), the model will show a maximal response around the presented orientation with the vector average of the population response will be the presented orientation.

To account for adaptation, the gain of the information channels is reduced in inverse proportion to their response by the previous stimulus (Supplementary Fig. 4B). For instance, if a 90° stimulus is the adapting stimulus, the sensitivity of the channels around 90° will be maximally reduced while orthogonal channels will be unaffected. The magnitude of this reduction (adaptation ratio) can be varied to allow for greater or less adaptation and was included as a free parameter in the analysis. The adaptation aspect of the model is consistent with previous models used to psychophysical data[27,28,33,73,74]. The new model accurately predicts serial dependency effects (where the current orientation is biased away from the previous orientation) seen in the neuronal data[34,35,75].

Prediction gain modulation works in a similar manner as adaptation except that the stimulus sensitivity, rather than channel sensitivity, is modulated. Furthermore, the gain modulation occurs before the stimulus and is for the orientation that is expected rather than presented. The modulation of stimulus sensitivity is consistent with a previous study which found that uncommon stimuli result in stimulus-specific adaptation in the auditory cortex[37]. Stimulus-specific adaptation has been used in modelling neuronal adaptation[36]. To model stimulus-selective gain modulation, the tuned Gaussian function was

found using Eq. 1 and inverted before being applied to the channels. The amount of gain modulation by expectation was a free parameter (expectation gain).

To account for long-lasting effects of gain modulation, the channel's sensitivity was normalised by the maximum sensitivity of response on each trial. This causes the model to have adaptation and expectation effects based on the presented orientation of at least four stimuli back. How many n-back stimulus affect the current trials sensitivity is determined by the modulation factor. We used this type of long-lasting gain to account for well-known effects such as serial dependency-like which can occur with adaptation and prediction[34,35]. We regressed the adaptation-only model against the neuronal data and found a factor of 3.0 best fit the data which was set for other modelling experiments.

To determine the effects of the different stimulus conditions (Random, Expected and Unexpected) on the model's channels, we presented sequences of orientations to the model and split the responses into conditions. To allow for easier comparison, we aligned the six orientation channels to their preferred orientation and collapsed the results across conditions. The same effects were evident before collapsing across the channels.

Lastly, we examined how the actual neuronal responses could be predicted by the model's predictions with different values of the free parameters. To do this, we used to model to predict responses to the orientations presented to the mice during the session for all stimulus conditions. For each neuron, we used the model's responses to the stimuli as regressors to predict the neuron's response (averaged from 250 to 1000 ms) for each stimulus condition. We iterated this procedure with different values for adaptation and expectation gain to determine what values best predicted the data.

### Reporting summary
Further information on research design is available in the Nature Portfolio Reporting Summary linked to this article.

## Data availability
The data are available at: https://osf.io/t2vb3.

## Code availability
The code for the analysis has been published in an open-access format (doi: 10.5281/zenodo.7444479)[76]. This is available at: https://github.com/MatthewFTang/PredictionOrientationSelectivityMouseV1.

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

## Acknowledgements

The Australian Research Council (ARC) provided support with a DECRA fellowship (DE210100508) to M.F.T., a DECRA fellowship (DE229199691) to C.L., the Centre of Excellence for Integrative Brain Function grant (CE140100007) to J.B.M. and E.A. and an ARC Discovery Project grant (DP1701009) to E.A. The National Health and Medical Research Council (NHMRC) provided support with a Project Grant to M.F.T., J.B.M. and E.A. (GNT1165337), an Ideas Grant to E.K., C.L. and E.A. (GNT1181643) and an NHMRC Investigator Grant (GNT2010141) to J.B.M. NVIDIA corporation donated a TITAN V GPU to M.F.T., and the Canadian Institute for Advanced Research (CIFAR) provided support to J.B.M.

## Author contributions

M.F.T. and E.A. conceived the experiments. E.K. and C.L. performed the experiments. M.F.T. and E.A. analysed the data. J.E.D. and M.F.T. developed the model. M.F.T., E.K., C.Y.L., J.E.D., J.B.M. and E.A. wrote the paper.

## Competing interests

The authors declare no competing interests.
