## [Peer Review File · Nature Communications]

Expectation violations enhance neuronal encoding of sensory information in mouse primary visual cortexReviewers' Comments:

Reviewer #1:

Remarks to the Author:

In this study, Tang and colleagues investigate how sensory coding in V1 is affected by adaptation and expectation. The authors find that responses to expected stimuli are reduced. They conclude that this effect is present in both awake and anesthetized recordings. Using a model, the authors show that in their paradigm neural responses were best characterized by a combination of adaptation and prediction. Finally, they present results from psychophysics experiment where they show that in human subjects, perception is biased away from the expected orientation.

The paradigm used to assess how sensory coding is affected by adaptation and prediction is well designed, and the findings interesting. However, the care given to the manuscript is far below what we would expect. We recommend publication in principle, but we will not review this manuscript again. It was just too fraught with errors, typos, and inconsistencies.

This review was written in collaboration with postdocs in the lab.

1. Currently, the authors pitch the manuscript as a "increase in coding efficiency" without ever defining what they mean (see also specific comments below). This is somewhat problematic as the authors never define what they mean, or even go into any specific analysis of the question. We would suggest to simply change the pitch.

2. An omission in the manuscript, the authors could investigate, is the fact that an unexpected stimulus in the authors paradigm is always a combination of the presentation of an unexpected stimulus combined with the absence of an expected stimulus. These two signals are separate, and are referred to as positive and negative prediction errors (see e.g.

<https://pubmed.ncbi.nlm.nih.gov/10195184/> or <https://pubmed.ncbi.nlm.nih.gov/30359606/>). The authors could analyze their data to see how neurons tuned to the expected stimulus respond to the absence of the expected stimulus, or whether there is a separate set of neurons that respond to the absence of the expected stimulus.

Specific

3. Line 47: "To cope with this information overload, and to enhance the efficiency of sensory processing, neuronal circuits use strategies such as adaptation, attention and prediction" – Here and elsewhere (e.g. in the title), what do the authors mean by efficiency? The argument about 'efficiency of sensory coding' that forms the title of the paper and also appears in the abstract, is completely unvisited in the manuscript, leaving us wonder why the authors make the claim.

4. Line 60. This sentence has some grammar problem.

5. Line 63: "However, non-invasive neuroimaging techniques reflect overall population activity and it is unclear how sensory representations are affected by expectation at the single neuron level." This is a bit convoluted, yes with non-invasive measurements, single neuron resolution is difficult, but there are numerous single neuron studies looking at this using invasive methods.

6. Line 68-69. It would be worth elaborating how activity in one layer of predictive hierarchy, that cancels neuronal response to an expected stimuli at a layer closer to the sensory periphery, allows for more efficient stimulus encoding. Given that – at first approximation - every bottom up spike cancelled by a top-down spike, requires said to down spike, this scheme is not more 'efficient' in the sense of minimum number of spikes required.

7. Line 87-88. And Methods. The authors need to make an effort to add more details into the Methods section. Wrt the 2-photon imaging, did the authors do chronic recording of 1693 neurons from the 5 mice over 23 sessions, or are the 1693 neurons pooled over sessions for these 5 mice. What constituted a session? How were the sessions spaced in time? What was the order of 'Random', 'Rotating', and 'anesthetized' sessions (we realize figure 1A lower panel has a schematic, but it's worthwhile adding to main text and methods)? How did the authors localize V1, using injection coordinates or did they do intrinsic mapping?

8. Line 91. 0.0034 c/deg should probably read 0.034 c/deg or similar. 0.0034 c/deg would be one cycle in approx. 300 degrees.

9. Line 93. "White noise stimulation" is probably a misnomer here? We assume the authors call it so, simply because the sequence of gratings is random, and not because they are showing white noise stimuli. Would remove or rephrase.
10. Line 117. Width is FWHM? Please clarify.
11. Line 123. Was it 4-9 presentations or 5-9 presentations (as stated in the rest of the text)?
12. Analysis windows are different in the awake and anesthetized analysis (250-1000 ms vs. 0-1000 ms). Please rectify this. Also, the awake data on Figure 2B should be represented in the same way as the anesthetized data on Figure 4A.
13. While we comment the authors' efforts to include psychophysics data, this should either be done to the normal experimental standards or removed. Two of three of the authors' subjects in these experiments were not blind to the aim of the experiment.
14. Figure 1B – The authors describe a color-coding system that does not correspond to the current version of the panel. Additionally, the authors state in the text: "in the Rotating condition the grating rotated either clockwise or anti-clockwise for 5 to 9 presentations", however, this is contradicted in the figure panel where three or four blue dots precede a green dot (assuming this corresponds to the random presentation) suggesting that there are often less than five presentations within the predicted grating sequence.
15. Figure 1D and E- It would also be helpful if in E the authors would provide the OSI for each given neuron.
16. Figure 1F. The peak at OSI 1, we assume is a consequence of the authors saturating the index by clamping the lowest response to 0. The authors should clarify in the figure legend. This is not normal procedure.
17. Figure 2C. Is missing a y-axis.
18. Figure 2D. Figure legend says 133 neurons are significantly modulated by expectation; Line 166 puts this number at 226. While figure 2E legend puts the number at 263. What exclusion criteria are the authors employing for the different analysis?
19. Figure 2E. "Black horizontal line indicates conditions are statistically different determined using non-parametric cluster-corrected procedures (see Methods)." We could not find a description for this procedure in the methods.
20. Figure 2E. The analysis needs description in methods – unclear what the authors did here.
21. Line 130. ...'becomes evident from ~85-100ms after stimulus onset'. Impossible to judge from the data shown.
22. Line 131. Sentence is incomplete.
23. Line 165. Figure 2C should read Figure 2D.
24. Line 173. This should probably be Figure 2F not 2E.
25. Line 175 (and elsewhere). What is the $t(\text{XXX})=\text{XXX}$ notation?
26. Line 187. $n=1954$; 23 sessions, vs. Line 87-88. $N=5$; 23 sessions ; $n=1693$?
27. Figure 3B. What are beta weights?
28. Figure 3G. Please explain the structure of the decoding accuracy preceding zero in the expected condition. (We understand why it might be non-zero before time zero, but why does it oscillate? And why is the peak right before zero higher?)
29. Figure 3 needs to be much more elaborated than the authors have cared to provide. Panels E-G are not even mentioned in the text.
30. Figure 4A – We assume the responses for the optimal grating start before zero because they are not regression-to-the-mean corrected? (i.e. the authors should select neurons on half their data (e.g. even trials), and plot on the other half (e.g. odd trials) – if that is not the problem, please explain).
31. Line 239. If under isoflurane anaesthesia 273/577 neurons, about half, show orientation-selective response, then this is almost twice what the authors find in awake state (462/1693), where they only have about a quarter of the neurons selective, see line 126. We find it puzzling why the authors conclude half is less than a quarter.
32. Line 264 "There was a slightly, but non statistically significant, larger Surprise effect in awake animals than in those that had been anesthetized ($t(733)=1.35$, $p = 0.18$). This suggests individual V1 neurons' orientation selective responses to violations of expectation are modulated by conscious state." – please remove this. We assume this is a 'typo' - the sequence of statements appears to

- suggest that the authors don't believe in the validity of statistical testing.
33. Line 326. It would appear there is at least one 'unexpected' too many in this sentence.
 34. It appears there is something severely amiss with the figures or figure legends. We suspect legends and figures are not version matched. Figure 1B – rotating: there are no black or red dots in the figure. Figure 2D: there are no green dots. Figure 2F: p is given as <0.05 and in the text as <0.001 – while of course not contradictory, please make consistent. Figure 3C: what is 'trial Z' etc. Figure 4D: the y axis label is likely incorrect. Figure 4A: the legend seems to be from a different version of this panel. Figure 4B: legend is missing. Figure 4C: there are no vertical lines. Figure 5A (text Line 283): there are no green dots. Figure 5A (text Line 283): there is no red line in Figure 5A. Figure S1. Legend is missing. Etc. etc. etc.
 35. Figures: Labeling of y axis for Gain or Width should be consistent. Now we can find: Gain (%dF/F) vs amplitude, width (°) vs standard deviation (deg).
 36. Multiple figure panels are never mentioned within the main text.
 37. Line 362. Typo.
 38. Line 541. ...approaches 'to' examine
 39. Line 563. ... the electrode data ◊ ... the imaged data?
 40. Line 573. ◊ ...using the weights in (4)...
 41. Line 595. While the manuscript is indeed properly skewered at this point, we assume the authors mean 'skew'.
 42. Line 262. Should read Figure 4E
 43. Line 263. slightly should read slight.
 44. Line 285. This should probably read "common" stimuli ...
 45. Line 333. Is this Figure 5G?
 46. Line 363. The reasoning for the authors prediction is unclear. Do the authors imply that because the activity in lower-levels of prediction hierarchy is inhibited by top-down input for an expected stimulus, the individual now perceives an expected stimulus less well than if it were randomly presented? We would assume that the predictive cancelling of activity at a lower-level in predictive hierarchy for an expected stimulus implies that the individual is already perceiving it.
 47. Line 368-370. We don't follow the argument. Can the authors rephrase?
 48. Line 205. Should this read (~10)?
 49. Line 257. Should probably read "Panels C-E".
 50. Line 246. Should this read Figure 4D?
 51. At some point we simply stopped – there are more errors and typos.

Reviewer #2:
Remarks to the Author:

I enjoyed reading this comprehensive and technically impressive study of expectation violation responses in the visual cortex of mice. I thought that your motivation and description of the experiments was convincing – and you have clearly put an enormous amount of work into this study. There are a lot of moving parts but I think they all hang together. The only issue, that I think needs to be addressed, is the relationship of your model of predictive coding to more formal accounts in the predictive coding literature. Otherwise, I have a few minor comments that that may improve the clarity of your presentation. Perhaps you could consider the following:

Major points

The major point is the conceptual relationship between the way that you have modelled expectation violation responses and more formal accounts based upon predictive coding. I suggest you add the following to the discussion to relate your findings to the body of work on precision-weighted prediction errors in the human neuroscience literature.

"Our model of expectation violation responses is phenomenological, in the sense that it describes our results in a way that is grounded in neuronal and synaptic mechanisms. This model contrasts with more formal accounts based upon hierarchical predictive coding. Generally, neuronal responses to violations of expectations are formulated as precision-weighted prediction errors (Clark 2013, Auksztulewicz and Friston 2015, Ainley, Apps et al. 2016, Shipp 2016, Limanowski 2017, Haarsma, Fletcher et al. 2018, Sterzer, Adams et al. 2018, Palmer, Auksztulewicz et al. 2019). In other words, neuronal responses reflect the difference between sensory afferents and top-down predictions that are modulated or weighted by precision. Precision, in this setting, is a prediction of predictability, as opposed to prediction of the sensory input. In the setting of precision-weighted prediction errors, we can associate the adaptation gain with the effects of predictability (i.e., precision weighting) and the excitation gain with the prediction error per se; namely, the disinhibition of stimulus bound responses by absent top-down predictions. This fits comfortably with predictive coding accounts of the mismatch negativity, where the equivalent effects are sometimes discussed in terms of stimulus-specific adaptation and a sensory memory component — that correspond to predictions of precision and stimuli, respectively (your Garrido et al reference)."

Minor points

In relation to the above distinction between a boost in gain or precision and a disinhibition due to the absence of a veridical prediction, I would change the abstract (line 39) to say:

"response to unexpected visual stimuli through a disinhibition or boost in gain,"

Similarly, in the introduction, I would say: "we demonstrate that unexpected stimuli elude inhibition by top-down predictions to manifest as a high again in neuronal tuning to the preferred stimulus."

On line 132, I think there is a missing word: perhaps: "modulation of neural activity began before stimulus evoked responses (0 ms)"

Line 152: do you mean "purple" dots, as opposed to "green" dots?

Online is 160 and 171, could you replace "effect of prediction" with "effect of predictability".

Line 204: please replace "this effect emerged in" with "this effect emerged with"

Line 241, please replace "prediction errors" with "violations of predictions"

Similarly, on line 259, please replace "the effect of prediction error" with "the effect of violating expectations"

in the legend to figure 5 (line 295) there are only three panels in the current version of your figure. Furthermore, there is no black vertical line in panel C. When you talk about yellow lines and pink lines, I assume you are referring to panel B.

Line 327, did you mean "unexpected trials" or "expected trials"?

Line 390: please replace "we found" with "we identified"

Line 449: change "sensory response" to "sensory responses"

Line 610: please replace "reduced by an inverse" with "reduced in inverse proportion"

Line 630: please replace "normalised by towards one (maximum sensitivity)" with "normalised by maximum sensitivity on each trial." This bit was not very clear and would be usefully packed with a sentence of explanation.

I hope that these comments help if any revision is required.

Ainley, V., M. A. Apps, A. Fotopoulou and M. Tsakiris (2016). "'Bodily precision': a predictive coding account of individual differences in interoceptive accuracy." *Philos Trans R Soc Lond B Biol Sci* 371(1708).

Auksztulewicz, R. and K. Friston (2015). "Attentional Enhancement of Auditory Mismatch Responses: a DCM/MEG Study." *Cereb Cortex* 25(11): 4273-4283.

Clark, A. (2013). "The many faces of precision (Replies to commentaries on "Whatever next? Neural prediction, situated agents, and the future of cognitive science")." *Front Psychol* 4: 270.

Haarsma, J., P. C. Fletcher, H. Ziauddeen, T. J. Spencer, K. M. J. Diederer and G. K. Murray (2018). "Precision weighting of cortical unsigned prediction errors is mediated by dopamine and benefits learning." *bioRxiv*.

Limanowski, J. (2017). (Dis-)Attending to the Body. *Philosophy and Predictive Processing*. T. K. Metzinger and W. Wiese. Frankfurt am Main, MIND Group.

Palmer, C. E., R. Auksztulewicz, S. Ondobaka and J. M. Kilner (2019). "Sensorimotor beta power reflects the precision-weighting afforded to sensory prediction errors." *Neuroimage* 200: 59-71.

Shipp, S. (2016). "Neural Elements for Predictive Coding." *Front Psychol* 7: 1792.

Sterzer, P., R. A. Adams, P. Fletcher, C. Frith, S. M. Lawrie, L. Muckli, P. Petrovic, P. Uhlhaas, M. Voss and P. R. Corlett (2018). "The Predictive Coding Account of Psychosis." *Biological psychiatry* 84(9): 634-643.

Response to reviewers

We are grateful for the reviewers' detailed and constructive feedback.

At the outset, we would like to apologise for the inconsistencies in the way the figures were referenced in the text and described in the figure legends. This was due to an error in which we inadvertently uploaded an earlier version of the paper during the online submission process. We have undertaken a thorough proof-reading of the revised manuscript.

In an effort to implement all the reviewers' suggestions, we have refined the framing of the narrative around predictive coding and its putative role in modulating the responses of visual cortex neurons to random and unpredicted stimuli. Specifically, we have changed the title, re-arranged the Results section, removed the small human psychophysics experiment and replaced it with a new analysis suggested by Reviewer 1. We have added step-by-step schematics to clearly explain the modelling and decoding sections, and have added numerous details missing from the original submission. We are confident that addressing the suggestions raised by the reviewers has resulted in a clearer and more cohesive manuscript. Details of the revisions are explained in our point-by-point responses below.

In the text that follows, reviewer comments are shown **in bold** and our responses to them are provided immediately below in normal font. For convenience, we have reproduced major changes in the manuscript text, verbatim, **in red font**.

Reviewer #1 (Remarks to the Author):

1. Currently, the authors pitch the manuscript as a “increase in coding efficiency” without ever defining what they mean (see also specific comments below). This is somewhat problematic as the authors never define what they mean, or even go into any specific analysis of the question. We would suggest to simply change the pitch.

We have now revised the framing of the manuscript and better clarified the overarching questions. We had originally used “coding efficiency” in the Introduction to describe the fidelity with which single neurons or neuronal populations represent sensory inputs, and how factors such as adaptation, attention and prediction can enhance the fidelity of the neural code. We have now replaced “coding efficiency” throughout the manuscript with better-defined concepts to frame the questions more clearly. Specifically, we now use phrases such as “transmission of sensory information,” “decoding of population activity by an ideal observer” and “characterization of response properties such as the amplitude and width of the neuronal tuning function.”

For example, the Introduction now reads:

Line 45. There is often more information in the sensory environment than the brain has the capacity to fully process. To cope with this information overload, activity within neuronal circuits is modulated by processes such as adaptation^{1,2} attention^{3,4}, and prediction^{5,6}. Neural adaptation is known to improve the transmission of sensory information in circuits by accounting for the statistics of past sensory inputs^{1,7,8}. Likewise, selective attention can enhance neural responses to task-relevant features and suppress irrelevant information^{3,9}. An influential theory of neural function argues that predictions about specific future stimuli, based upon Bayesian inference, might similarly improve the fidelity of stimulus representations^{5,6}. Based on this *predictive coding* view, the mammalian cortex is conceptualised as a predictive machine that uses the statistical regularities of incoming sensory inputs to iteratively generate an internal model of its external environment.

Line 74. We characterised how prediction affects orientation selectivity in V1 neurons, and how changes in orientation tuning modulate the amount of information about the sensory input carried by individual neurons and neuronal populations. We demonstrate that unexpected stimuli significantly increase the gain of orientation selectivity without any corresponding changes to the width of the tuning function. Such increased gain to expectation violations yields increased information about stimulus features within single-cells and at the level of neuronal populations. This enhanced representation of unexpected stimuli is present in both awake and anaesthetised mice. Finally, we use a computational model to quantify the contribution of adaptation and expectation to neuronal responses at the single trial level.

Results Line 187. In our initial set of analyses, we found that expectation affected orientation selectivity in individual V1 neurons. We next examined how enhanced orientation selectivity for unexpected stimuli at the single-neuron level in turn shaped the information contained within the population response.

2. An omission in the manuscript, the authors could investigate, is the fact that an unexpected stimulus in the authors paradigm is always a combination of the presentation of an unexpected stimulus combined with the absence of an expected stimulus. These two signals are separate, and are referred to as positive and negative prediction errors (see e.g. <https://pubmed.ncbi.nlm.nih.gov/10195184/> or <https://pubmed.ncbi.nlm.nih.gov/30359606/>). The authors could analyze their data to see how neurons tuned to the expected stimulus respond to the absence of the expected stimulus, or whether there is a separate set of neurons that respond to the absence of the expected stimulus.

Thank you for this suggestion. We have implemented this analysis in the revised manuscript. The results are shown in the new Figure 4, reproduced below. The analysis tests whether, and in what way, expectation of an orientation affects neurons specifically tuned to that orientation. To do this, we grouped the recorded neurons based on their orientation preference and characterised the changes in

tuning separately for each group. Neurons tuned to the expected orientation showed the greatest reduction in selectivity relative to when non-preferred orientations were expected. These results are summarized in Figure 4 and in Supplementary Figure 2 (shown below). The new analysis has been combined with the population-level analysis examining the difference between expected and presented orientations in the original manuscript (now Figure 4E-F). Collectively, these results suggest that predictions primarily target the representations of expected stimuli. This could increase coding efficiency (i.e., the same representation, but with fewer spikes) if a given higher level (top-down) area targets multiple lower-level areas.

We also briefly examined your second suggestion, about whether there are neurons that only fire when there is an unexpected stimulus. We did this by selecting neurons that showed a significant stimulus-evoked response to unexpected stimuli, but no reliable response to expected stimuli. Only a very small proportion of neurons fit this profile (less than 5%). Given these inconclusive findings, we have not included the results of this analysis in the revised manuscript.

Figure 4. Increase in neuronal responses to unexpected stimuli is determined by the magnitude of the prediction error. **(A)** Neurons tuned to each displayed orientation are affected differently when different orientations are expected. Panel A shows an example for each expected orientation using neurons selective for 90° gratings, as defined based on their responses in the Random condition (from 250 – 1000 ms).

Responses of remaining neurons selective for the other presented orientations are shown in Supplementary Figure 2. For each unexpected stimulus in the rotating condition, we identified the difference between the orientation of the expected stimulus and the orientation of the presented unexpected stimulus. For instance, if 60° was expected but 0° was unexpectedly presented, the expectation violation would be 60° . **(B)** All orientation-selective neurons aligned with their preferred orientation, plotted as separate Gaussians for each difference between the expected orientation and the presented orientation (expectation violation). **(C)** Gain of Gaussians fitted to each neuron's response, plotted as a function of expectation violation for all orientation-selective neurons. **(D)** Baseline of Gaussians fitted to each neuron's response. **(E)** Forward encoding modelling reveals how population representations of orientation are affected by the degree of expectation violation. The encoding weights are shown separately here for different values of expectation violation. **(F)** The y axis shows the difference between the presented and decoded orientation (Δ Perceived orientation). The population response (filled symbols) is biased away from the expected orientation with the largest bias at $\pm 30^\circ$. In all panels, error bars indicate ± 1 standard error of the mean across permutations. * indicates $p < 0.05$.

Supplementary Figure 2. Expectations affect neurons differently depending on their preferred orientation. Each panel shows neurons tuned to different orientations, as defined by their stimulus-evoked responses in the Random condition. The different colour-coded curves show different expected orientations. Neurons show the largest decrease in response when their preferred orientation is similar to the expected orientation. Across all panels error bars indicate ± 1 standard error of mean.

Specific

3. Line 47: “To cope with this information overload, and to enhance the efficiency of sensory processing, neuronal circuits use strategies such as adaptation, attention and prediction” – Here and elsewhere (e.g. in the title), what do the authors mean by efficiency? The argument about ‘efficiency of sensory coding’ that forms the title of the paper and also appears in the

abstract, is completely unvisited in the manuscript, leaving us wonder why the authors make the claim.

We have now revised the framing of the manuscript and clarified the overarching questions. We had originally used “coding efficiency” in the Introduction to describe the fidelity with which single neurons or neuronal populations represent (encode) sensory inputs, and how processes such as adaptation, selective attention and prediction might modulate the neural code. We have now replaced “coding efficiency” throughout the manuscript (and in the title) with better-defined terms to frame the questions more clearly. Specifically, we have used phrases such as “transmission of sensory information,” “decoding of population activity by an ideal observer” and “amplitude and width of the neuronal tuning function.” For further information refer to our response to Comment 1 above.

4. Line 60. This sentence has some grammar problem.

We have corrected the sentence, Line 60:

Decoding of activity from electroencephalography (EEG) recordings in humans has revealed that expectation affects the representation of visual information in the brain^{14–18}.

5. Line 63: “However, non-invasive neuroimaging techniques reflect overall population activity and it is unclear how sensory representations are affected by expectation at the single neuron level.” This is a bit convoluted, yes with non-invasive measurements, single neuron resolution is difficult, but there are numerous single neuron studies looking at this using invasive methods.

This sentence has been removed in the revision.

6. Line 68-69. It would be worth elaborating how activity in one layer of predictive hierarchy, that cancels neuronal response to an expected stimuli at a layer closer to the sensory periphery, allows for more efficient stimulus encoding. Given that – at first approximation - every bottom up spike cancelled by a top-down spike, requires said to down spike, this scheme is not more ‘efficient’ in the sense of minimum number of spikes required.

This is an interesting observation. Here we reflected a theoretical point that is commonly stated in the predictive coding literature (Rao and Ballard, 1999, *Nat Neuro*; Friston, 2005, *Proc Royal Soc*). On this account, one way in which efficiency might be achieved would be if activity in a small number of higher-order top-down projecting neurons can reduce the activity of a relatively large number of low-level sensory neurons. Nevertheless, we agree the argument for spiking efficiency contains assumptions about neuronal mechanisms which are currently unknown and are not central to the current study. We have therefore removed the reference to “efficient”.

7. Line 87-88. And Methods. The authors need to make an effort to add more details into the Methods section. Wrt the 2-photon imaging, did the authors do chronic recording of 1693 neurons from the 5 mice over 23 sessions, or are the 1693 neurons pooled over sessions for these 5 mice. What constituted a session? How were the sessions spaced in time? What was the order of ‘Random’, ‘Rotating’, and ‘anesthetized’ sessions (we realize figure 1A lower panel has a schematic, but it’s worthwhile adding to main text and methods)? How did the authors localize V1, using injection coordinates or did they do intrinsic mapping?

We have now clarified all these points.

We recorded 1693 neurons from three mice. (Three other mice were used for anesthetised recordings; hence, a total of 6 mice were used across the study as a whole.) For the awake paradigm, there were 23 recording sessions, each of which lasted between 1.5 and 2 hours, across the five mice. Data from all neurons were pooled. We have clarified this in the text, Line 584.

In 3 mice, we ran a total of 23 imaging sessions and collected data from 1697 neurons. Neurons from all sessions and mice were pooled for analysis.

V1 was localized anatomically. We have clarified this in the revision, Lines 543.

V1 was localised anatomically using coordinates established using functional methods. Injections were centred 2mm lateral and 4.5mm posterior to Bregma.

A “session” was a recording in which we ran all combinations of the stimulus paradigm. We counter-balanced the order of conditions across the mice. We have clarified this in the revision, Lines 585.

One session (1.5 – 2 hours) was recorded in a day from each mouse. These sessions occurred between 1 and 4 times per week. In each session, two runs of *Rotating* and *Random* sequences were presented, and each of these contained 1800 trials, alternating between *Rotating* and *Random*. The order of sequences was counter-balanced across mice.

8. Line 91. 0.0034 c/deg should probably read 0.034 c/deg or similar. 0.0034 c/deg would be one cycle in approx. 300 degrees.

Thank you for noting this typo. It has been corrected to 0.034 c/deg in the revision, Line 90.

9. Line 93. “White noise stimulation” is probably a misnomer here? We assume the authors call it so, simply because the sequence of gratings is random, and not because they are showing white noise stimuli. Would remove or rephrase.

We used the term “white noise” to refer to the absence of any temporal correlation across stimuli. We have now removed the term to avoid confusion.

10. Line 117. Width is FWHM? Please clarify.

Width here refers to the standard deviation (sigma) of the Gaussian. This has been clarified in multiple places in the revision.

11. Line 123. Was it 4-9 presentations or 5-9 presentations (as stated in the rest of the text)?

It was 5-9 presentations. This has been corrected in the revision.

12. Analysis windows are different in the awake and anesthetized analysis (250-1000 ms vs. 0-1000 ms). Please rectify this. Also, the awake data on Figure 2B should be represent in the same way as the anesthetized data on Figure 4A.

We have now used the same window of analysis for the anesthetized and awake data, and have adopted a consistent method for visualization of the results. The new figure for the anesthetized results is shown below (Figure 6 in the revised manuscript).

Figure 6. Expectations affect the gain of orientation-selective V1 neurons under anaesthesia. **(A)** Time courses for all orientation-selective neurons ($N = 273$) aligned to their preferred orientation to allow averaging. Shading indicates ± 1 standard error of the mean across neurons. **(B)** Population orientation tuning for the three expectation conditions, averaged across an epoch from 250 to 1,000 ms after stimulus presentation. Solid lines are fitted Gaussian functions with a constant offset. **(C)** Summary statistics for the gain of the fitted Gaussians in B. **(D)** Width of the fitted Gaussians in B. **(E)** Comparison of the “surprise” effect (Unexpected events minus Random events) between awake and anaesthetised animals. **(F)** Time course of responses to the preferred orientation of each neuron, shown separately for the three conditions. Neurons have been sorted by their responses in the Unexpected condition. Across panels A-E shading and error bars indicate ± 1 standard error of the mean across neurons.

13. While we comment the authors efforts to include psychophysics data, this should either be done to the normal experimental standards or removed. Two

of three of the authors subjects in these experiments were not blind to the aim of the experiment.

We have now removed the psychophysics data and replaced them with the new analysis of the mouse data in Figure 4 as per reviewer's suggestion. This change has improved the logical flow of the manuscript.

14. Figure 1B – The authors describe a color-coding system that does not correspond to current version of the panel. Additionally, the authors state in the text: “in the Rotating condition the grating rotated either clockwise or anti-clockwise for 5 to 9 presentations”, however, this is contradicted in the figure panel where three or four blue dots precede a green dot (assuming this corresponds to the random presentation) suggesting that there are often less than five presentations within the predicted grating sequence.

This is now corrected, Figure 1B.

15. Figure 1D and E- It would also be helpful if in E the authors would provide the OSI for each given neuron.

16. Figure 1F. The peak at OSI 1, we assume is a consequence of the authors saturating the index by clamping the lowest response to 0. The authors should clarify in the figure legend. This is not normal procedure.

We have removed OSI from the revision as it was not used as part of the narrative of the paper or referred to in the main text.

17. Figure 2C. Is missing a y-axis.

We have added the neuron numbers to this panel in the revision.

18. Figure 2D. figure legend says 133 neurons are significantly modulated by expectation; Line 166 puts this number at 226. While figure 2E legend puts the number at 263. What exclusion criteria are the authors employing for the different analysis?

We have corrected the different neuron numbers in the revision. The same exclusion criteria now are applied to all analyses so neuronal numbers are consistent.

19. Figure 2E. “Black horizontal line indicates conditions are statistically different determined using non-parametric cluster-corrected procedures (see Methods).” We could not find a description for this procedure in the methods.

We have added a statistics section to the revision explaining the procedure (see Lines 673).

Non-parametric signed permutation tests^{71,72} were used to determine time resolved differences between conditions. The sign of the data was randomly flipped ($N = 5,000$), with equal probability, to create a null distribution. Cluster-

based permutation testing was used to correct for multiple comparisons over the timeseries, with a cluster-form threshold of $p < 0.05$ and significance threshold of $p < 0.05$.

20. Figure 2E. The analysis needs description in methods – unclear what the authors did here.

Thank you for noting this omission. We have added a description of this analysis to the revision. We have referenced the toolbox we used to calculate the circular mean, Line 610.

To provide another test of how prediction affects orientation selectivity of individual neurons, we found the circular mean⁶⁴ of the averaged orientation tuning curve across all presentations within the condition (Figure 2E). This was done for each time point (1/sample rate) between -500 and 2000 ms around stimulus presentation.

21. Line 130. ...'becomes evident from ~85-100ms after stimulus onset'. Impossible to judge from the data shown.

We have removed this sentence from the revision.

22. Line 131. Sentence is incomplete.

We have corrected this in the revision.

During presentation of the *Expected* stimulus (red trace), modulation of neuronal activity began before the onset of the stimulus (0 ms).

23. Line 165. Figure 2C should read Figure 2D.

Corrected.

24. Line 173. This should probably be Figure 2F not 2E.

Corrected.

25. Line 175 (and elsewhere). What is the t(XXX)=XXX notation?

Thank you for noting this, we have made appropriate corrections in the revision.

26. Line 187. n=1954 ; 23 sessions, vs. Line 87-88. N=5 ; 23 sessions ; n=1693 ?

These numbers have been corrected.

27. Figure 3B. What are beta weights?

This is a regression coefficient. This has been clarified in the revision.

28. Figure 3G. Please explain the structure of the decoding accuracy preceding zero in the expected condition. (We understand why it might be non-zero before time zero, but why does it oscillate? And why is the peak right before zero higher?)

For Expected trials, the immediately preceding stimuli were generally highly correlated with the current stimulus being decoded. This explains why decoding accuracy is higher than chance before stimulus onset in the Expected condition. With respect to the observed oscillations in the output of the decoder, we believe these are likely reflect a combination of 3 factors: (1) oscillations in neuronal activity due to the periodic onsets of stimuli in the presented sequences; (2) the 30° changes in orientation from one stimulus to the next within the rotating sequences; and (3) the dynamics of the calcium indicator. Figure 2A and B provide an example of the dynamics of the raw dF/F data for both individual neurons and across the population. We speculate that at the peaks of dF/F the large variability supports the strongest decoding, whereas when dF/F is close to zero there is no variability for the decoder to use to find a linear solution. Although the peak just prior to zero has a larger absolute value, there is overlap in the error bars. For these reasons, the conclusions from the decoding analysis are focused on the comparison of Random and Unexpected transitions, for which there is no correlation with the preceding stimulus. We have merely included the Expected decoding curves here for completeness. We have added the following in the revision to clarify these issues, Line 210.

Unsurprisingly, in the Expected condition orientation information could be decoded above chance before the stimulus appeared. This is because orientations occurring before stimulus presentation (0 ms) were correlated with the orientation of the decoded stimulus presented at time zero. The decoding profile for Expected stimuli also exhibits an oscillating profile, which likely reflects a combination of three factors: oscillations in neuronal activity due to the periodic onsets of stimuli in the presented sequences; the 30° changes in orientation from one stimulus to the next within the rotating sequences; and the dynamics of the calcium indicator.

29. Figure 3 needs to much more elaboration that the authors have cared to provide. Panels E-G are not even mentioned in the text.

We apologise for the lack of clarity for Figure 3. Panels E and G were discussed in the text but without being explicitly referenced. We have corrected this in the revised manuscript, and have also included text discussing panel G.

30. Figure 4A – We assume the responses for the optimal grating start before zero because they are not regression-to-the-mean corrected? (i.e. the authors

should select neurons on half their data (e.g. even trials), and plot on the other half (e.g. odd trials) – if that is not the problem, please explain).

The neurons were sorted by their preferred orientation in the *Unexpected* condition, and this sorting was then applied to the other conditions. The same sorting on the Random and Expected conditions strongly suggest this effect is not a selection problem. Furthermore, there is a strong correlation of preferred orientations between the conditions, again suggesting that these results are not a result of regression-to-the-mean issues.

31. Line 239. If under isoflurane anaesthesia 273/577 neurons, about half, show orientation-selective response, then this is almost twice what the authors find in awake state (462/1693), where they only about a quarter of the neurons selective, see line 126. We find it puzzling why the authors conclude half is less than a quarter.

This statement has now been removed.

32. Line 264 “There was a slightly, but non statistically significant, larger Surprise effect in awake animals than in those that had been anesthetized ($t(733)=1.35$, $p = 0.18$). This suggests individual V1 neurons’ orientation selective responses to violations of expectation are modulated by conscious state.” – please remove this. We assume this is a ‘typo’ - the sequence of statements appears to suggest that the authors don’t believe in the validity of statistical testing.

We have now emphasised that the prediction effects were seen in both anesthetized and awake animals, and we have removed the statement about the relative magnitudes of this effect.

33. Line 326. It would appear there is at least one ‘unexpected’ too many in this sentence.

This has been corrected.

34. It appears there is something severely amiss with the figures or figure legends. We suspect legends and figures are not version matched. Figure 1B – rotating: there are no black or red dots in the figure. Figure 2D: there are no green dots. Figure 2F: p is given as <0.05 and in the text as <0.001 – while of course not contradictory, please make consistent. Figure 3C: what is ‘trial Z’ etc. Figure 4D: the y axis label is likely incorrect. Figure 4A: the legend seems to be from a different version of this panel. Figure 4B: legend is missing. Figure 4C: there are no vertical lines. Figure 5A (text Line 283): there are no green dots. Figure 5A (text Line 283): there is no red line in Figure 5A. Figure S1. Legend is missing. Etc. etc. etc.

Figure 2C. Legend has been changed to blue dots jumping to green dots.

Figure 2F. This was done intentionally. We used a single asterisk throughout the figures to indicate statistical significance of $p < 0.05$ for simplicity. In the text, we report the exact p values where possible (i.e., when greater than 0.001), in line with the style guidelines.

Figure 2D. Legend has been corrected.

Figure 3C. We have removed this notation of trials with numbers (Figure 3).

Figure 4D was correct. We used the sigma (standard deviation) throughout to measure the width of tuning. We have changed this label to “width” in the revision for consistency with the other figures (now Figure 6D).

Figure 4C. There are in fact vertical lines in the figure. The error bars are just very small.

Line 583 has been removed.

35. Figures: Labeling of y axis for Gain or Width should be consistent. Now we can find: Gain (%dF/F) vs amplitude, width (°) vs standard deviation (deg).

This has been corrected throughout the manuscript.

36. Multiple figure panels are never mentioned within the main text.

All figures are now referenced in text.

37. Line 362. Typo.

This is now removed.

38. Line 541. ...approaches ‘to’ examine

Corrected.

39. Line 563. ... the electrode data à ... the imaged data?

Corrected.

40. Line 573. à ...using the weights in (4)...

Corrected.

41. Line 595. While the manuscript is indeed properly skewered at this point, we assume the authors mean ‘skew’.

Corrected.

42. Line 262. Should read Figure 4E

Corrected.

43. Line 263. slightly should read slight.

This section has been removed.

44. Line 285. This should probably read “common” stimuli ...

This has been corrected. We have clarified this in the text.

45. Line 333. Is this Figure 5G?

Corrected.

46. Line 363. The reasoning for the authors prediction is unclear. Do the authors imply that because the activity in lower-levels of prediction hierarchy is inhibited by top-down input for an expected stimulus, the individual now perceives an expected stimulus less well than if it were randomly presented? We would assume that the predictive cancelling of activity at a lower-level in predictive hierarchy for an expected stimulus implies that the individual is already perceiving it.

This psychophysics section has now been removed. To clarify, the low-level representation is often considered to be ‘pre-conscious’ (viz. Dehaene’s Global Workspace model) so a changed low-level representation can still alter perception.

47. Line 368-370. We don’t follow the argument. Can the authors rephrase?

This section has been removed.

48. Line 205. Should this read (~10)?

This has been corrected.

49. Line 257. Should probably read “Panels C-E”.

This was the correct label. There are error bars in panel B. The reviewer might not have noticed them because they are small.

50. Line 246. Should this read Figure 4D?

Corrected.

51. At some point we simply stopped – there are more errors and typos.

We have carefully proofread the manuscript.

Reviewer 2

I enjoyed reading this comprehensive and technically impressive study of expectation violation responses in the visual cortex of mice. I thought that your motivation and description of the experiments was convincing – and you have clearly put an enormous amount of work into this study. There are a lot of moving parts but I think they all hang together. The only issue, that I think needs to be addressed, is the relationship of your model of predictive coding to more formal accounts in the predictive coding literature. Otherwise, I have a few minor comments that that may improve the clarity of your presentation. Perhaps you could consider the following:

Major points

The major point is the conceptual relationship between the way that you have modelled expectation violation responses and more formal accounts based upon predictive coding. I suggest you add the following to the discussion to relate your findings to the body of work on precision-weighted prediction errors in the human neuroscience literature.

"Our model of expectation violation responses is phenomenological, in the sense that it describes our results in a way that is grounded in neuronal and synaptic mechanisms. This model contrasts with more formal accounts based upon hierarchical predictive coding. Generally, neuronal responses to violations of expectations are formulated as precision-weighted prediction errors (Clark 2013, Auksztulewicz and Friston 2015, Ainley, Apps et al. 2016, Shipp 2016, Limanowski 2017, Haarsma, Fletcher et al. 2018, Sterzer, Adams et al. 2018, Palmer, Auksztulewicz et al. 2019). In other words, neuronal responses reflect the difference between sensory afferents and top-down predictions that are modulated or weighted by precision. Precision, in this setting, is a prediction of predictability, as opposed to prediction of the sensory input. In the setting of precision-weighted prediction errors, we can associate the adaptation gain with the effects of predictability (i.e., precision weighting) and the excitation gain with the prediction error per se; namely, the disinhibition of stimulus bound responses by absent top-down predictions. This fits comfortably with predictive coding accounts of the mismatch negativity, where the equivalent effects are sometimes discussed in terms of stimulus-specific adaptation and a sensory memory component – that correspond to predictions of precision and stimuli, respectively (your Garrido et al reference)."

Thank you for this excellent suggestion, which indeed clarifies what this model is meant to show. As requested, we have incorporated this paragraph, more or less verbatim, into the Discussion, Line 474.

Our model of expectation-violation responses is phenomenological, in the sense that it describes our results in a way that is grounded in neuronal and synaptic mechanisms. This model contrasts with more formal accounts based upon hierarchical predictive coding. Generally, neuronal responses to violations of expectations are formulated as precision-weighted prediction errors^{45–52}. In other words, neuronal responses reflect the difference between sensory afferents and top-down predictions that are modulated or weighted by precision. Precision, in this context, is a prediction of predictability, as opposed to prediction of the sensory input. In the context of precision-weighted prediction errors, we can associate adaptation gain with the effects of predictability (i.e., precision weighting) and excitation gain with the prediction error per se; namely, the disinhibition of stimulus-bound responses by absent top-down predictions. This fits with predictive coding accounts of the mismatch negativity, where the equivalent effects are sometimes discussed in terms of stimulus-specific adaptation and a sensory memory component — which in turn correspond to predictions of precision and stimuli, respectively^{14,15}.

Minor points

In relation to the above distinction between a boost in gain or precision and a disinhibition due to the absence of a veridical prediction, I would change the abstract (line 39) to say:

"response to unexpected visual stimuli through a disinhibition or boost in gain,"

This sentence has been removed in the revision.

Similarly, in the introduction, I would say: "we demonstrate that unexpected stimuli elude inhibition by top-down predictions to manifest as a high again in neuronal tuning to the preferred stimulus."

Done, Line 74.

On line 132, I think there is a missing word: perhaps: "modulation of neural activity began before stimulus evoked responses (0 ms)"

Corrected, Line 132.

Line 152: do you mean “purple” dots, as opposed to “green” dots?

Corrected. We apologise for having uploaded the revised figures with an earlier version of the figure legends, Line 156.

Online is 160 and 171, could you replace "effect of prediction" with "effect of predictability".

Done, Line 166.

Line 204: please replace "this effect emerged in" with "this effect emerged with"

Done, Line 252.

Line 241, please replace "prediction errors" with "violations of predictions"

Done, Line 429.

Similarly, on line 259, please replace "the effect of prediction error" with "the effect of violating expectations"

This sentence has been removed in the revision.

in the legend to figure 5 (line 295) there are only three panels in the current version of your figure. Furthermore, there is no black vertical line in panel C. When you talk about yellow lines and pink lines, I assume you are referring to panel B.

Corrected, Line 340.

Line 327, did you mean "unexpected trials" or "expected trials"?

You are correct – this should read “expected trials”. Corrected, Line 381.

Line 390: please replace "we found" with "we identified"

Done, Line 290.

Line 449: change "sensory response" to "sensory responses"

Done, Line 511.

Line 610: please replace "reduced by an inverse" with "reduced in inverse proportion"

Done, Line 692.

Line 630: please replace "normalised by towards one (maximum sensitivity)" with "normalised by maximum sensitivity on each trial." This bit was not very clear and would be usefully packed with a sentence of explanation.

Done. We have explained this section more clearly now (Line 708):

To account for long-lasting effects of gain modulation, the channel's sensitivity was normalised by the maximum sensitivity of response on each trial. This causes the model to have adaptation and expectation effects based on the presented orientation of at least four stimuli back. How many n-back stimulus affect the current trials sensitivity is determined by the *modulation factor*. We used this type of long-lasting gain to account for well-known effects such as serial dependency-like which can occur with adaptation and prediction^{34,35}.

I hope that these comments help if any revision is required.

Thank you, your comments were extremely helpful.

Reviewers' Comments:

Reviewer #1:

Remarks to the Author:

Before we go any further with the review process, it appears like there may be a critical problem with the authors' analysis and consequently conclusions. One of the central findings of the paper is that response to an oriented grating that violates expectation is higher in magnitude compared to the same when it is expected or randomly presented (Figure 2), even while anaesthetized (Figure 6).

Based on the authors' reply to our comment 30. and on Figure 2B (the response to the unexpected grating at the preferred orientation starts before time 0), we are led to believe that the authors select orientation-responsive neurons based on responses to unexpected gratings in the Rotating condition (or possible use just the unexpected condition to define preferred orientation). An essential consequence of this analysis choice is that now plotting the response of thus selected neurons to expected grating or random grating is bound to be lower than that for unexpected grating, simply due to a regression-to-the-mean effect. Thus, for instance, Figure 2F, could be a trivial consequence of this regression-to-the-mean. Unfortunately, the methods are still not clear enough to decipher how the authors did this analysis exactly, thus we are not entirely sure whether our assessment is correct. However, if the authors indeed perform the analysis as described above (as is suggested in the response to our comment 30.), then we fear the entirety of the conclusions may be invalid.

The authors should verify that response to unexpected grating is not spuriously higher than to expected grating due to this analysis artifact. The authors could do this by pooling neurons responsive to oriented grating in any of the three conditions (Random, Rotation-expected, Rotating-unexpected), and comparing the average response of thus selected neurons across the three conditions. Or a similar strategy, that does not suffer from regression to the mean.

Given this has the potential of affecting the key conclusion of the paper, we would ask the authors to fix this before we go into the specific replies to our comments.

Reviewer #2:

Remarks to the Author:

Many thanks for responding to my previous suggestions. The current version reads nicely.

Response to reviewer

Reviewer 1

Based on the authors' reply to our comment 30. and on Figure 2B (the response to the unexpected grating at the preferred orientation starts before time 0), we are led to believe that the authors select orientation-responsive neurons based on responses to unexpected gratings in the Rotating condition (or possible use just the unexpected condition to define preferred orientation). An essential consequence of this analysis choice is that now plotting the response of thus selected neurons to expected grating or random grating is bound to be lower than that for unexpected grating, simply due to a regression-to-the-mean effect. Thus, for instance, Figure 2F, could be a trivial consequence of this regression-to-the-mean. Unfortunately, the methods are still not clear enough to decipher how the authors did this analysis exactly, thus we are not entirely sure whether our assessment is correct. However, if the authors indeed perform the analysis as described above (as is suggested in the response to our comment 30.), then we fear the entirety of the conclusions may be invalid.

The authors should verify that response to unexpected grating is not spuriously higher than to expected grating due to this analysis artifact. The authors could do this by pooling neurons responsive to oriented grating in any of the three conditions (Random, Rotation-expected, Rotating-unexpected), and comparing the average response of thus selected neurons across the three conditions. Or a similar strategy, that does not suffer from regression to the mean.

Regression to the mean is not an issue here. Orientation-responsive neurons were indeed selected by pooling neurons exhibiting orientation selectivity in either *Random* or *Unexpected* conditions. We did not use the responses in the *Expected* condition, as these are subjected to systematic pre-stimulus adaptation effects due to the rotational nature of the paradigm. Below we illustrate the magnitude of the main effect based on 3 types of selection: (A) the combined approach as is currently reported in the manuscript, (B) using only the *Random* condition and (C) using only the *Unexpected* condition. The small effect of regression to the mean is illustrated by comparing the subpanels. When neurons are selected on *Unexpected* trials, the t-statistic comparing responses in the *Random* and *Unexpected* trials was $t = 16.70$, $p < 0.001$, whereas in the *Combined* approach it was $t = 15.74$, $p < 0.001$. The result remains practically unchanged regardless of the selection group, indicating the robustness of the main effect. Please note that our response to your point 30 referred to the "sorting" of neurons for display in Figure 2C and not to the selection of the orientation-responsive neurons used for the analysis.

Reviewers' Comments:

Reviewer #1:

Remarks to the Author:

To be frank, our confidence in the veracity of the authors' results has not exactly been bolstered by the new version of the manuscript, or the authors' response to the regression to the mean concern. We do agree, that if the authors select orientation selective neurons on the combined "unexpected" and "random" data, the comparison between the two is valid (see however, points 1.-5. below). Given the authors' renewed lack of attention to detail the manuscript advertises, in spite of having been – likely irritatingly – alerted to this point in the first round of reviews, I would refrain from recommending publishing this work. This is mainly because a large portion of the effect appears to be acausal – see point 5 below, and the overall frequency of inaccuracies in the manuscript make it difficult for me to put any faith in the authors' conclusions. We raised the concern in point 5 already in the previous round of reviews, but the authors appear to have forgotten to address this question. But to reiterate this (I realized that I already said this in round 1 and appear to have forgotten about it, please excuse), I will not look at this manuscript again.

This review was written in collaboration with postdocs in the lab, and is not accompanied by confidential comments to the editor.

1. The expected condition should be removed from Figure 2F and all similar comparisons, unless the authors can demonstrate clearly that this is not simply the consequence of regression to the mean.
2. Please make the method for how cells are selected clearer and describe it in the results section.
3. Moreover, we are not sure we understand why the authors cannot use the expected condition to select orientation selective neurons – the plot in Figure 2B (right), would suggest that tuning curves in random and expected look very similar. Given that the authors have not done the analysis in the last response, we assume it contradicts the authors conclusions.
4. P values appear inconsistent in the response and the manuscript (compare response figure A and Figure 2F in the manuscript, which we assume are the same figure?).
5. The authors should explain why the unexpected presentation starts increasing before time zero in Figure 2B (at 0, i.e. preferred orientation), if not for regression to the mean.
6. Figure 6F. It is unclear how the "surprise effect" is calculated. Figure legend says: "unexpected minus random", text says "random minus unexpected" (line 448ff), and neither would appear to result in values between 0 and 1.
7. Figure 6B. Has an extra label "90".
8. Figure 6C-E are missing stats.
9. How is Figure 6B consistent with 6F? At 0 (preferred in Figure 6B, expected is larger than random), while in 6F, random looks considerably larger than expected? Please explain.
10. Regarding our previous comment 7 – the response we fear is not intelligible. Three mice that were not used for anesthetized recordings, but five mice for the awake paradigm? And is it 1693 neurons, or 1697? Anyway – we hope the authors have devoted more attention to detail in the manuscript.

"We recorded 1693 neurons from three mice. (Three other mice were used for anesthetized recordings; hence, a total of 6 mice were used across the study as a whole.) For the awake paradigm, there were 23 recording sessions, each of which lasted between 1.5 and 2 hours, across the five mice. Data from all neurons were pooled. We have clarified this in the text, Line 584. "

"In 3 mice, we ran a total of 23 imaging sessions and collected data from 1697 neurons. Neurons from all sessions and mice were pooled for analysis."

11. Figure 3E. The red line above appears to be missing.
12. Line 543. Spurious "60".
13. Urethane anesthesia details are not in the methods.
14. How do the authors interpret the finding that changing the analysis window from 0-1s, to .25-1s

results in a change of responsive neurons from 273 to 96? Something appears amiss.
15. Etc.

Point-by-point response

Reviewers 1

1. The expected condition should be removed from Figure 2F and all similar comparisons, unless the authors can demonstrate clearly that this is not simply the consequence of regression to the mean.

We included the results for the *Expected* condition in Figure 2 for completeness, and because we believe it would be scientifically misleading to arbitrarily omit one of the three main conditions of the study. Moreover, in our view the results of the *Expected* condition are likely to be of interest to many readers. As we explain clearly in the manuscript (paragraph starting Line 128), the formal statistical comparisons of interest involve the matched *Random* and *Unexpected* conditions. None of the conclusions of the manuscript are based upon comparisons with the *Expected* condition. This is because, by definition, the stimulus history is fully determined in the *Expected* condition leading to a systematic response profile that starts before the onset of the stimulus of interest. This is not a confound, but an essential and intrinsic element of the experimental design, namely, to establish a statistical prediction in the stimulus sequence that could then be violated in the *Unexpected* condition. The term ‘regression to the mean’ is inaccurate in this context; please see our response to point 5 below.

2. Please make the method for how cells are selected clearer and describe it in the results section.

This is already clearly stated in the manuscript. Neurons were selected for the analysis if they exhibited an orientation-selective response as confirmed by a one-way ANOVA.

Line 103: In line with previous work, many imaged neurons showed orientation selectivity for the spatial frequency employed (462/1697; one-way ANOVA $p < 0.05$ for orientation selectivity).

Line 172: The responses of 133/462 orientation-selective neurons (28.8%) were significantly modulated in the *Unexpected* condition relative to the *Random* condition (t-test, $p < 0.05$).

3. Moreover, we are not sure we understand why the authors cannot use the expected condition to select orientation selective neurons – the plot in Figure 2B (right), would suggest that tuning curves in random and expected look very similar. Given that the authors have not done the analysis in the last response, we assume it contradicts the authors conclusions.

As outlined in our earlier responses, we did not use the *Expected* condition to select for orientation-selective neurons because the rotating sequences of gratings used to establish the prediction were, by design, correlated (i.e., clockwise or counter-clockwise rotations in 30-degree increments), and thus neuronal responses to each new stimulus are unlikely to be statistically independent. Nevertheless, at the Reviewer's request we implemented this suggestion in the second round of review. Specifically, we selected neurons on every possible combination of conditions (including *Expected* trials) and the conclusions of the paper remained unchanged.

4. P values appear inconsistent in the response and the manuscript (compare response figure A and Figure 2F in the manuscript, which we assume are the same figure?).

For consistency, throughout the manuscript we used a single star to indicate p values less than 0.05. The exact p value level is reported in text for all tests. In our previous response letter, we used three stars to indicate to the Reviewer that under all selection criteria the results were highly significant. Critically, we have not used different numbers of stars to express different levels of statistical significance in the manuscript. Therefore, as the number of stars used is correctly defined in the manuscript and unrelated to the revision letter, these figures are not inconsistent.

5. The authors should explain why the unexpected presentation starts increasing before time zero in Figure 2B (at 0, i.e. preferred orientation), if not for regression to the mean.

The rise is simply due to the Gaussian smoothing applied to the data to reduce extrinsic noise in the calcium signal. This is a standard approach (Kubler, et al., 2021, IEEE; Deneux, et al., 2017, Nature Communications; Romano, et al., 2017, PLOS Computational Biology) for calcium imaging pre-processing to increase the signal-to-noise ratio, in which data points are averaged around each time point and weighted by the symmetrical Gaussian normal distribution with a certain width. At the stimulus onset, the smoothing will therefore incorporate data from just before and just after stimulus onset. It is not surprising the rise is more visible for the *Unexpected* condition because this condition produced the highest amplitude evoked responses. Most importantly, however, the relevant pre-stimulus rise is not statistically significant.

Contrary to the Reviewer's suggestion, the effect is not due to regression to the mean, which refers to the notion that if a given sample of a random variable is "extreme", the next sample of that variable will likely be closer to the mean of the overall sample. To take a classic example, if a parent is particularly tall it is statistically more likely their offspring will be shorter than they are rather than taller. Returning to our study, we see that our findings are in fact the opposite of what would be predicted on the basis of a regression to the mean account. Specifically, the *Unexpected* condition produced by far the largest neuronal responses despite the fact that neurons were selected based on their (smaller) responses in the *Random* condition.

6. Figure 6F. It is unclear how the "surprise effect" is calculated. Figure legend says: "unexpected minus random", text says "random minus unexpected" (line 448ff), and neither would appear to result in values between 0 and 1.

We have clarified the calculation in the revised text, which states “for each neuron we calculated the “surprise” effect by subtracting the gain of the Gaussian tuning curve for the *Unexpected* condition from that of the *Random* condition”.

Values for this subtraction do not necessarily have to fall between 0 and 1, and we do not state or imply this in the text. As it happens, the y-axis in Figure 6E spans values between 0 and 0.8, but the range of scores for individual neurons could go beyond this range (scores ranged between 0.0027 and 3.2).

7. Figure 6B. Has an extra label “90”.

This label is correct. Although the final data point falls at 60 degrees, we have included a 90-degree label as is the convention when plotting orientation. Furthermore, this is consistent with other figures in the manuscript showing orientation (e.g., Fig 2A, 2B, 3A, 3B, 3C, 4A, 4B, 4E).

8. Figure 6C-E are missing stats.

This statistical test was reported in the text. We have now also added lines to the panels.

9. How is Figure 6B consistent with 6F? At 0 (preferred in Figure 6B, expected is larger than random), while in 6F, random looks considerably larger than expected? Please explain.

The colours of the conditions in panel B were accidentally switched during the previous round of changes requested by this reviewer. We apologise for this mistake. This is now fixed.

10. Regarding our previous comment 7 – the response we fear is not intelligible. Three mice that were not used for anesthetized recordings, but five mice for the awake paradigm? And is it 1693 neurons, or 1697? Anyway – we hope the authors have devoted more attention to detail in the manuscript.

“We recorded 1693 neurons from three mice. (Three other mice were used for anesthetized recordings; hence, a total of 6 mice were used across the study as a whole.) For the awake paradigm, there were 23 recording sessions, each of which lasted between 1.5 and 2 hours, across the five mice. Data from all neurons were pooled. We have clarified this in the text, Line 584. “

“In 3 mice, we ran a total of 23 imaging sessions and collected data from 1697 neurons. Neurons from all sessions and mice were pooled for analysis.”

This confusion was caused because 1 mouse was used for both awake and anaesthetized experiments. We apologise for this confusion and have clarified this in the revision.

A total of 5 wild type mice (C57BL) were used for this study; 2 only awake, 2 only anaesthetized, 1 in both awake and anaesthetized.

11. Figure 3E. The red line above appears to be missing.

The figure legend said the line was red when it was blue. This has been corrected in the revision.

12. Line 543. Spurious "60".

This number is not spurious, it refers to a citation that was not superscripted. This is now corrected in the manuscript.

13. Urethane anesthesia details are not in the methods.

We apologise for this omission. We have included this in the revised version. Page 23.

Before each recording session, the mouse was anesthetized by intraperitoneal administration of urethane/chlorprothixene (0.8 g/kg and 5 mg/kg body weight, respectively). All other methodological details were identical to those described for the awake recordings.

14. How do the authors interpret the finding that changing the analysis window from 0-1s, to .25-1s results in a change of responsive neurons from 273 to 96? Something appears amiss.

The window was not the only change. In the initial round of reviews, we were asked to apply consistent criteria for the analyses of data from the awake and anaesthetized experiments. This meant that the analysis window was changed along with the statistical tests used for selection. For statistics, we moved from bootstrapping each orientation to a single one-way ANOVA across the orientations. This yielded 462/1697 orientation-selective neurons in the awake experiments and 96/576 in the anaesthetized experiments. The ratio of statistically significant orientation-selective cells was 27% in awake experiments and 16% in anaesthetized experiments. These proportions of orientation selective responses are consistent with those reported in the published literature on mouse visual cortex (e.g. Andermann, et al., Neuron, 2011; Jeyabalaratnam, et al., PLOS One, 2013).